# Hierarchical aggregation deconstruction search for vehicle routing problems

## Abstract

Recent progress in neural combinatorial optimization has shown promise for vehicle routing problems (VRPs). Iterative improvement frameworks address the limitations of pure construction policies, which often struggle with exploration and large-scale performance, but they remain constrained by solution encodings that ignore the hierarchical structure of routes. We introduce Hierarchical Aggregation Deconstruction Search (HADES), a neural improvement method with distance-aware, anisometric positional encodings tailored to routing solutions. HADES incorporates two complementary components: an in-route positional encoding, which captures the circular and non-uniform ordering of nodes within tours, and a cross-route encoding, which represents route membership and structural relations across tours. This hierarchical design provides solution representations better aligned with the anisometric and head-tail connected nature of VRPs, leading to more effective deconstruction. Extensive experiments across multiple VRP variants demonstrate that our model consistently advances the state of the art, with particularly strong gains on large-scale benchmarks. We will make our source code publicly available to foster future research.

## 1 Introduction

The vehicle routing problem (VRP) is one of the most fundamental combinatorial optimization (CO) problems, with wide applications in transportation, logistics, and supply chain management. Solving VRP efficiently can lead to substantial cost savings and operational improvements in real-world systems (Toth & Vigo, 2014; Jayarathna et al., 2022; Fernando et al., 2024). However, due to its NP-hard nature, VRP and its variants remain a long-standing challenge for both operations research (OR) and machine learning communities. Most traditional approaches have primarily relied on handcrafted heuristics and metaheuristics (Helsgaun, 2017; Wouda et al., 2024). These methods can deliver relatively high-quality solutions on a smaller scale, but they typically require significant expert knowledge for algorithm design and extensive parameter tuning. Moreover, their computational cost often scales poorly with problem size, and they must be redesigned or substantially modified when new problem variants or constraints are introduced (Vidal et al., 2020; Liu et al., 2023).

Recent advances in neural combinatorial optimization (NCO), particularly with the reinforcement learning (RL), have been employed to effectively solve various types of CO problems (Bengio et al., 2021). One common setting for RL is to construct solutions step-by-step through sequential decision-making, showing generalization and adaptability across VRP variants.

Despite years of progress, these neural construction methods still face two major limitations: (i) their exploration ability is restricted to single-shot generation and cannot exploit solution structures, and (ii) they often struggle to match the performance of state-of-the-art OR solvers for large-scale instances (Bogyrbayeva et al., 2024; Wu et al., 2024).

In parallel to the development of construction-based methods, a complementary line of research has focused on improvement methods, which iteratively refine an initial solution rather than generating one

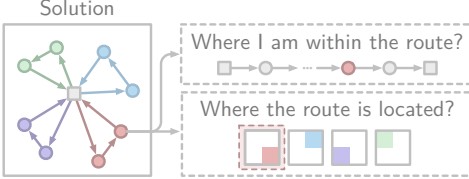

Figure 1: Unlike constructive policies, an improvement policy has to reason about hierarchical positional information of a solution.

from scratch. Inspired by classical paradigms such as deconstruct-and-repair (Shaw, 1998), these methods explore new neighborhoods by selectively removing parts of a solution and reconstructing them. Compared to purely constructive approaches, improvement methods offer a stronger ability to explore the solution space, as they leverage existing high-quality solutions as starting points and perform targeted modifications to achieve further gains (Hottung et al., 2025b), e.g., by iterative deconstruct-and-repair search in the solution space.

Despite their success, most improvement methods still inherit a key limitation from construction-based approaches: the way they represent and understand solutions. Existing models primarily focus on the geographical coordinates of nodes, but neglect the *hierarchical structure* of routing solutions. Unlike construction methods, which only need to determine where a customer should be placed, improvement methods must also reason about how each customer is embedded within the current solution. This requires understanding both the route membership of a customer and its relative position within that route.

Our core hypothesis is that positional information of in-route and cross-route ordering is crucial for making informed and effective refinement decisions in a deconstruction-search framework. Our intuition stems from large language models (LLMs), where positional encoding techniques are crucial for capturing the order of tokens in sequences (Vaswani et al., 2017; Su et al., 2024). However, directly applying LLM-style positional encodings to VRPs would be inadequate because solution structures are *anisometric*: unlike text, where token distances are inherently uniform, the distance between two customers in a route is not constant and does not follow isometric assumptions. Moreover, a single positional encoding would be insufficient: the model should capture not only the order of customers within a route, but also which route they belong to and how routes relate to one another.

In this paper, we introduce the Hierarchical Aggregation Deconstruction Search (HADES), a novel deconstrcuction framework with distance-aware, anisometric positional encodings. HADES explicitly models anisometric hierarchical route positional information with (i) in-route positional encoding, which captures the order and relative spacing of nodes within a route, and (ii) cross-route positional encoding, which identifies the structural positional information of each route and its interaction with other routes. Together, these hierarchical embeddings provide crucial information for representing solutions, enabling effective deconstruction actions.

Our contributions are summarized as follows:

- We propose HADES, a novel deconstruction framework for VRP that integrates hierarchical, anisometric positional encodings into solution representations.
- We design two complementary encoding schemes: a distance-indexed, head-tail consistent in-route encoding and a depot-anchored angular cross-route encoding. To our knowledge, this is the first NCO model to explicitly encode hierarchical anisometry in VRP solutions.
- We demonstrate through extensive experiments across multiple VRP variants and scales that HADES outperforms strong competitive OR solvers and existing neural deconstruction methods, achieving state-of-the-art performance.

## 2 RELATED WORK

**Construction methods.** Neural construction approaches generate complete solutions in an autoregressive fashion. Pointer Networks initiated this line of work for routing, first with supervised learning (Vinyals et al., 2015) and then with reinforcement learning (RL) (Bello et al., 2016). For VRPs, early models learn to decode customer sequences directly (Nazari et al., 2018; Kool et al., 2019). To better exploit permutation symmetries, Kwon et al. (2020); Kim et al. (2022) introduce symmetric training/decoding schemes. Inference-time search has been layered on top of construction to close quality gaps: efficient active search updates a subset of parameters at test time (Hottung et al., 2022), while search-guided beam search steers decoding with Monte Carlo lookahead (Choo et al., 2022). Another thread strengthens generalization by re-encoding the remaining subproblem after each partial decision (Drakulic et al., 2024; Luo et al., 2023). To increase sampling diversity, several works learn sets of policies rather than a single policy (Grinsztajn et al., 2024; Hottung et al., 2025a). Beyond autoregression, non-autoregressive methods predict edge/arc "heat maps" that are assembled into tours with post-hoc search (Kool et al., 2022; Sun & Yang, 2023; Li et al., 2025b). A line of works extends to rich and more practical variants (Zhou et al., 2024; Berto et al., 2025a;b;c;

Hua et al., 2025; Li et al., 2025a). While construction models are fast and scalable, their solution quality typically lags behind strong traditional heuristics.

**Improvement methods.** Improvement approaches iteratively refine an initial solution via local or large-neighborhood modifications, often yielding higher final quality than one-shot construction. Learning-to-improve frameworks train policies to propose targeted edits on top of a current solution (Wu et al., 2019; Ma et al., 2021; Gast Zepeda et al., 2025). Several works guide classical $k$-opt for routing (Wu et al., 2019; da Costa et al., 2020), and recent variants relax the fixed-$k$ constraint to allow flexible neighborhoods (Ma et al., 2023). Hybrid schemes integrate learned components with metaheuristics to expand the search horizon, including large neighborhood search (Hottung & Tierney, 2020) and ant colony optimization (Ye et al., 2024b; Kim et al., 2025). To address scale, divide-and-conquer frameworks decompose instances and combine constructive and search modules (Li et al., 2021; Ye et al., 2024c; Zheng et al., 2024; Ouyang et al., 2025). Most relevant to our setting, NDS (Hottung et al., 2025b) embeds a learned deconstruction policy within a large-neighborhood search framework and reports competitive results on VRPs.

Despite these advances, most improvement frameworks encode solutions as flat sequences or employ simple aggregation strategies (i.e., averaging route embeddings), unable to capture the hierarchical structure of routes (circular, head-tail connected order within tours and cross-route relations), leading to poor performance and suboptimal learned representations. Building on intuition on how token order is treated in large language models, HADES addresses the critical gap of improvement methods by designing ad-hoc positional encodings for VRPs. We design embeddings that are distance-aware and anisometric with complementary in-route and cross-route components capable of capturing complex solutions in a neural deconstruction-based improvement framework, yielding state-of-the-art results.

## 3 PRELIMINARY

**Problem Definition** A VRP is defined on a graph $G = (V, E)$, where each node $x_i \in V$ denotes a customer and each edge $e_{i,j} \in E$ corresponds to traveling from $x_i$ to $x_j$ with an associated cost (e.g. distance between $x_i$ and $x_j$). All routes originate from and terminate at a depot node $x_0$. In the Capacitated VRP (CVRP), vehicles of capacity $C$ are dispatched such that the total demand $d_i$ on any route does not exceed $C$, and each customer is served exactly once. The objective is to minimize the total travel distance. The VRP with Time Windows (VRPTW) extends this setting by associating each customer $i$ with a service time $s_i$ and an admissible interval within which service must begin, permitting early arrivals but prohibiting tardiness. The objective is to minimize the total travel time while respecting both vehicle capacity and time windows. The Prize-Collecting VRP (PCVRP) relaxes the requirement of serving all customers: each customer $i$ yields a profit $p_i$, and the objective becomes to minimize the total travel cost minus the sum of collected prizes. A more detailed problem formulation of VRPs is provided in the Section A.1.

**MDP Formulation** The deconstruction search can be naturally framed as a Markov Decision Process (MDP). We define the formulation as follows: the *state* represents the current solution and instance features; the *action space* consists of selecting one customer $x_m$ for removal at each step $m$ from $M$ total removals; the *policy* $\pi_\theta$ is parameterized by $\theta$ and conditioned on a random seed $v$, which helps diversify rollouts, with $v$ being a binary vector of dimension $d_v$ as introduced by Hottung et al. (2025a); the *action* at each step is sampled from the distribution $\pi_\theta(x_m|l, \mathbf{s}, v, x_{1:m-1})$, where $l$ is the problem instance, $\mathbf{s}$ is the current solution, $v$ is the random seed, and $x_{1:m-1}$ are the previously chosen actions; the *reward* is computed based on the improvement in the objective function after the removal of the selected customer, incentivizing better solutions. A more detailed MDP formulation is provided in the Section A.2.

## 4 METHODOLOGY

In this section, we present HADES as illustrated in Fig. 2. We introduce the key components for effective hierarchical aggregation: the *in-route* positional encoding and *cross-route* positional encoding, providing both the motivation behind these designs and the analysis that led to their devel-

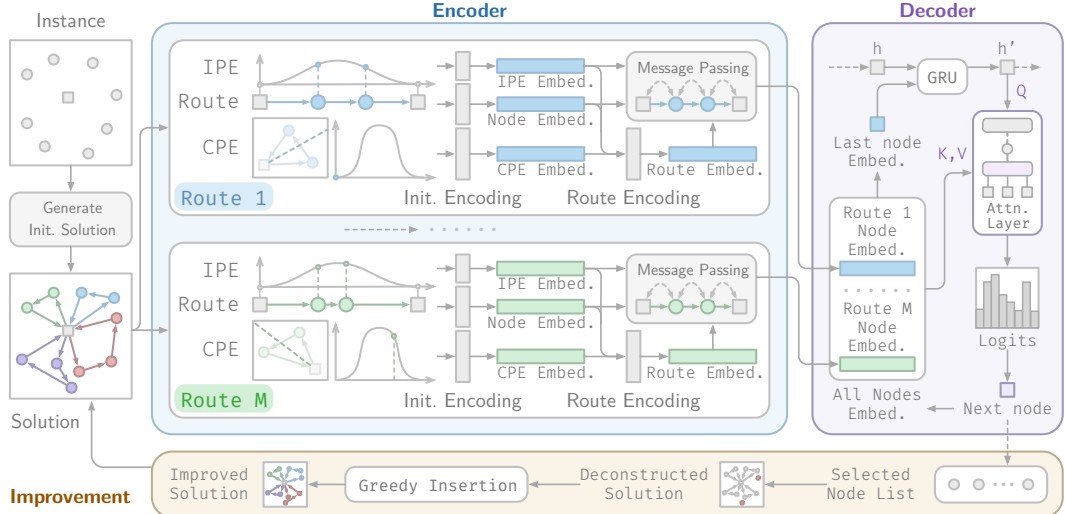

Figure 2: Overview of HADES. The framework follows an encoder-decoder architecture, where the encoder first embeds the instance using hierarchical positional encodings (IPE and CPE) to capture both node-level and route-level information. The decoder then sequentially selects nodes for removal using a rollout-based policy. This process iteratively refines the solution by deconstructing and reconstructing the route, improving the solution at each step. The final improved solution is obtained after multiple iterations.

opment. These components, along with the rollout decoder, deconstruction process, improvement strategy, and training scheme, form the complete framework as an effective deconstruction method.

## 4.1 ENCODER

We use a lightweight yet effective encoder that enhances standard node features with our hierarchical positional encodings, while ensuring that computation remains linear in the number of visited nodes. Let $x_i$ represent the raw feature of node $i$ (e.g, coordinates and demands); $\text{IPE}(\cdot)$ be the in-route positional encoding of a node; and $\text{CPE}(\cdot)$ the cross-route positional encoding of a route. The in-route positional encoding is injected directly as a node feature at the input, while both positional encodings contribute to the route embedding, which is then used to refine the node states.

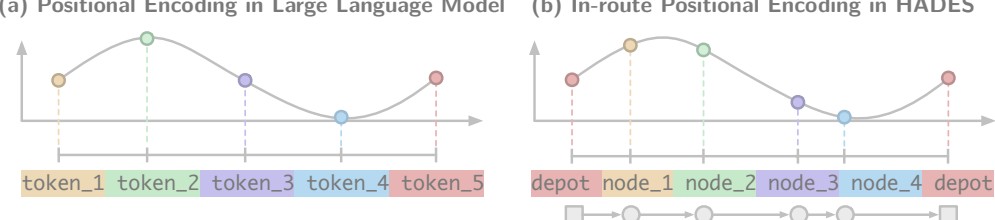

Figure 3: Comparison of positional encodings: (a) in LLMs, where token distances are equal, and (b) in our approach, where varying node distances result in anisometric positional encoding values.

**In-route Positional Encoding** In language models, positional encodings are indispensable for modeling sequential structure, where the index of a token is treated as an *isometric* position in the sequence (Vaswani et al., 2017). Since textual tokens lack an explicit notion of spatial distance, these encodings assume equal spacing between adjacent indices. In contrast, vehicle routing solutions exhibit a sequential topology while additionally providing explicit geometric distances between consecutive nodes as shown in Fig. 3. Consequently, directly reusing isometric positional encodings ignores the *anisometric* nature of routes, where the separation between nodes reflects heterogeneous travel costs rather than uniform steps. For more details, definition, and discussion about the anisometric, please refer to Section A.3.

Formally, for a route $r = (v_1, \ldots, v_L)$ with length $L$, here $v_i = 0, 1, \cdots, N$ is the index number for customers, specially, $v_0 = v_L = 0$ marks the index of the depot. For each node $x_i$ in this route, we

define its cumulative travel distance from the depot as $d_i = \sum_{j=2}^{i} \left\| x_{v_j} - x_{v_{j-1}} \right\|_2$, where $x_{v_j} \in \mathbb{R}^2$ is the coordinate of node $v_j$. This scalar $d_i$ naturally generalizes the notion of the positional index by embedding order into the geometry of the route.

We then construct sinusoidal embeddings using distance-aware indices. Specifically, for node $v_i$:

$$\text{IPE}(x_{v_i}, 2k) = \sin\left(\frac{d_i}{\lambda^{2k/D}}\right), \qquad \text{IPE}(x_{v_i}, 2k+1) = \cos\left(\frac{d_i}{\lambda^{2k/D}}\right),$$

where $D$ is the embedding dimension, $k$ is the frequency-band index, and $\lambda$ is a scaling constant. Unlike standard encodings, which typically map uniform indices to sinusoidal curves, our design uniquely maps cumulative travel distances, thereby encoding anisometric spacing between nodes.

A critical issue arises from scale. While textual sequences in language models can grow to thousands of tokens, VRP routes are bounded by the spatial extent of the instance. To address this discrepancy and normalize across varying instances while enforcing topological consistency, we rescale cumulative distances so that the entire tour length $d_L$ corresponds to a full sinusoidal period: $\hat{d}_i = \frac{d_i}{d_L} \cdot 2\pi$. This guarantees $\text{IPE}(v_1) = \text{IPE}(v_L)$, ensuring that the depot at the start and end of the tour shares the same embedding, consistent with the head-tail connected nature of routes.

This construction offers two benefits. First, it aligns embeddings with route topology: nodes early in the tour map to positive phases, while nodes approaching the return depot map to negative phases, naturally reflecting circularity. Second, it preserves distance-awareness: the embedding gap between two nodes grows with their physical separation, providing the model with geometrically meaningful positional signals for guiding deconstruction.

**Cross-route Positional Encoding**  Unlike text, where a single one-dimensional positional index suffices, VRP solutions are *hierarchical*: beyond within-route order, each route carries a global spatial position relative to the depot and to other routes. Empirically, near-optimal CVRP solutions partition the plane into depot-centered angular sectors, with each route visiting customers in a limited directional span; this reduces cross-route interleaving and long crisscross legs, thereby lowering the total travel cost.

First, we randomly select a reference route, $r^\star$, and set its orientation to zero. The direction of this reference route is defined by the average angle of nodes relative to the depot. For each route $r$, we compute the average angle of all the nodes within that route, based on the relative angles of each node from the depot. Specifically, for each node $v_i$ in route $r$, we compute its angle $\theta_{v_i}$ relative to the depot as $\theta_{v_i} = \text{atan2}(x_{v_i}^{(y)}, x_{v_i}^{(x)})$, where $x_{v_i}^{(x)}$ and $x_{v_i}^{(y)}$ are the respective coordinates. The route's overall angle $\phi_r$ in then the average of all such angles for the nodes within route $r$: $\phi_r = (1/L) \cdot \sum_{v_i \in r} \theta_{v_i}$, where $L$ is the length of the $r = (v_1, \dots, v_L)$.

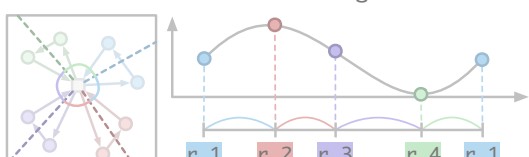

Figure 4: Illustration of the cross-route positional encoding in HADES. The figure shows how routes are assigned relative phases based on their angular separation from the depot, with the positional encoding reflecting their global orientation within the solution space.

To account for the angular offset of each route relative to the reference route, we calculate the wrapped relative angle $\delta_r$ for route $r$ with respect to the reference route $r^\star$: $\delta_r = \text{Mod}(\phi_r - \phi_{r^\star}, 2\pi)$, where the $\text{Mod}(\cdot, \cdot)$ function ensures the angle is within the range $[0, 2\pi)$.

We then map $\delta_r$ to a sinusoidal encoding, similar to the in-route positional encoding, to represent the phase of each route, for each route $r$, we compute the multi-frequency sinusoidal encoding:

$$\text{CPE}(r, 2k) = \sin\left(\frac{\delta_r}{\gamma^{2k/D}}\right), \qquad \text{CPE}(r, 2k+1) = \cos\left(\frac{\delta_r}{\gamma^{2k/D}}\right),$$

where $\gamma$ is a scaling factor controlling the frequency of the sinusoidal function. This mapping ensures that each route's phase is uniquely represented in a continuous sinusoidal space.

This design complements the IPE, creating a hierarchical two-level positional signal: the in-route phase captures the relative position within the tour, while the cross-route phase captures the route's position within the entire solution. The circular nature of the routes is preserved, with the phase

for a route at the beginning and end of the tour being identical, ensuring consistency. Additionally, routes with angular separations close to $\pi$ will have inversely flipped sinusoidal values (both sine and cosine), aligning with the depot-centered geometry.

**Feature Aggregation** We then project node features with an MLP block: $h_i = \text{FF}\big([x_i|\text{IPE}(x_i)]\big)$, sharing parameters across all nodes and routes. Given the circular tour order, let $\text{prev}(i)$ and $\text{next}(i)$ denote the immediate neighbors of $i$ on its route. We update nodes with a neighbor-aggregation block followed by a residual MLP and layer normalization:

$$h_i' \;=\; \text{Norm}\Big(h_i \;+\; \text{FF}\Big(\text{ReLU}\Big(W^3\big[\,h_i \mid W^1 h_{\text{prev}(i)} + W^2 h_{\text{next}(i)}\,\big]\Big)\Big)\Big),$$

where $\text{Norm}(\cdot)$ is the LayerNorm (Ba et al., 2016). For route $r = (v_1, \ldots, v_L)$, we compute a compact route code by pooling node embeddings and IPE, then fuse with CPE:

$$\bar{h}_r = \tfrac{1}{L} \sum_{v_i \in r} h_{v_i}', \quad \overline{\text{IPE}}_r = \tfrac{1}{L} \sum_{v_i \in r} \text{IPE}(x_{v_i}), \quad t_r \;=\; \text{FF}\big([\,\bar{h}_r | \overline{\text{IPE}}_r | \text{CPE}(r)]\big).$$

This $t_r$ is broadcast to nodes on route $r$ and used to refine their embeddings:

$$\hat{h}_i \;=\; \text{Norm}\big(h_i' \;+\; \text{FF}\big(\text{ReLU}\big(W^4[h_i' \mid t_r]\big)\big)\big), \quad \text{for } i \in r.$$

$\text{FF}(\cdot)$ denotes a two-layer MLP with ReLU; $W^{1:4}$ are learned matrices shared across routes. The IPE thus shapes node states locally (input features and pooling), while the CPE contributes through $t_r$, providing global, inter-route context. Two light self-attention blocks can be appended after $\hat{h}_i$ for additional refinement; we omit details as they follow standard practice.

## 4.2 DECODER

We employ a PolyNet-style pointer network decoder informed by prior decisions (Hottung et al., 2025a). At step $m$, a GRU is conditioned on the last removals and produces a query $q_m$ as introduced in the previous work (Hottung et al., 2025b); keys $K$ are linear projections of the encoder outputs $\mathbf{h} = \{\hat{h}_i\}$ which contain the hierarchical positional information. The probability of selecting node at step $m$ after masking infeasible or already removed ones ($\text{mask}_m$) is:

$$\pi_\theta(x_m|l, \mathbf{s}, v, x_{1:m-1}) = \text{softmax}\left(\frac{q_m K^\top}{\sqrt{d}} + \text{mask}_m\right), \tag{1}$$

as in widely used neural routing decoders (Kool et al., 2019; Kim et al., 2022).

## 4.3 IMPROVEMENT

At test time we embed the policy in the augmented simulated annealing (ASA) framework of Hottung et al. (2025b). For geometric augmentations (Kwon et al., 2020), each improvement step samples a list of routes as starting solutions $\mathbf{s}$, deconstructs by removing the chosen customers, repairs with the same randomized greedy routine for efficiency and simplicity, and evaluates the candidates $\mathbf{s}'$. Acceptance follows simulated annealing with temperature $\sigma$:

$$\Pr[\text{accept } \mathbf{s}' \mid \mathbf{s}] = \min\left(1, \; \exp\left(-\frac{\text{cost}(\mathbf{s}') - \text{cost}(\mathbf{s})}{\sigma}\right)\right), \tag{2}$$

where $\text{cost}(\cdot)$ calculates the cost of the solution. ASA decays $\sigma$ and performs thresholded exchanges across augmentations. The execution is fully batched on GPU, benefiting from massive parallelism.

## 4.4 TRAINING

We use winner-takes-all policy-gradient training as suggested by (Grinsztajn et al., 2024; Hottung et al., 2025a). For each instance and its augmentations, we sample multiple stochastic rollouts of $M$ removals, yielding repaired solutions $\{\mathbf{s}_i \mid i = 1, \ldots, M\}$ with returns $\text{cost}(\mathbf{s}_i)$ (negative cost). Let $i^\star = \arg\max_i \text{cost}(\mathbf{s}_i)$ be the best rollout. Only this winning trajectory backpropagates (others are stop-gradient). With a running baseline $b$,

$$\nabla_\theta J(\theta) \;\approx\; (R_{i^\star} - b) \sum_{i=1}^{M} \nabla_\theta \log \pi_\theta(\mathbf{s}_i \mid \mathcal{S}), \tag{3}$$

where $R_i = -\text{cost}(\mathbf{s}_i)$ is the reward, $\mathcal{S}$ is the solution space. Optionally adding entropy regularization only on the winner. Sampling, masking, and optimizer details follow Hottung et al. (2025b).

# 5 EXPERIMENTS

We evaluate the effectiveness of HADES through extensive experiments across three VRP variants and multiple instance sizes. We benchmark against strong learning-based policies and competitive operations-research solvers, reporting solution quality and runtime. In addition, we compare and analyze alternative positional encoding schemes, assess generalization under distribution shifts, and conduct ablation studies to isolate the contributions of our components.

## 5.1 EXPERIMENTAL SETUP

**Problems** We evaluate HADES on three canonical VRPs: (i) CVRP, with instances drawn from the uniform generator of Kool et al. (2019) and from the clustered/realistic generator of Queiroga et al. (2021); (ii) VRPTW, using the same location and demand settings as CVRP, with time windows generated following Solomon (1987); and (iii) PCVRP, again using the CVRP location/demand settings, with customer prizes sampled at random and scaled with demand to reflect service effort. Full specifications about generator hyperparameters are provided in Section B.1.

**Baselines** We benchmark HADES against strong OR heuristics, including HGS (Vidal, 2022), SISRs (Christiaens & Vanden Berghe, 2020), LKH3 (Helsgaun, 2017), PyVRP-HGS (Wouda et al., 2024), and NVIDIA cuOpt (NVIDIA Corporation, 2025). We also include state-of-the-art learning methods as baselines: BQ (Drakulic et al., 2024), LEHD (Luo et al., 2023), UDC (Zheng et al., 2024), NCO-LLM (Tran et al., 2025) and the state-of-the-art NDS (Hottung et al., 2025b). A more detailed introduction to these baselines and the available licenses is provided in Section B.2.

**Training Configuration** We train a separate policy for each problem and instance size ($N \in \{500, 1000, 2000\}$). For $N \leq 1000$ we train for 2000 epochs; for $N = 2000$ we resume from the $N = 1000$ checkpoint at epoch 1500 and train $+500$ epochs. Each epoch processes 150K instances; each instance runs 100 improvement iterations with 128 rollouts and 10 warm-up improvement steps. We fix the learning rate to $10^{-4}$ and remove 15 customers per deconstruction step. For more details of training configurations and hyperparameters, we refer to Section B.3.

**Testing Protocol** For a fair comparison on sequential solvers (HGS, PyVRP, SISRs, and HADES), we cap wall time per instance at $\{60, 120, 240\}$ seconds for each problem size. Batch-processing baselines (BQ, LEHD, NCO-LLM) receive an equivalent budget per batch. All methods utilize a single AMD Milan 7763 CPU core, with parallelism implemented on a single NVIDIA A100 GPU. CVRP test sets follow Drakulic et al. (2024) for $N{=}500$ (128 instances) and Ye et al. (2024c) for $N \in \{1000, 2000\}$ (100 instances each). For VRPTW and PCVRP, we generate new test sets with 250 instances for each size $N \in \{500, 1000, 2000\}$. For HADES, the ASA temperature starts at 0.1 and decays exponentially to 0.001; the threshold factor is 15. At each improvement step, we sample 200 rollouts per instance; each deconstructed solution is reconstructed $5\times$ (one DNN-guided order, $4\times$ random orders). We use 8 augmentations for the CVRP/VRPTW and 128 for the PCVRP.

## 5.2 RESULTS AND ANALYSIS

This section answers three questions that anchor our contribution: (i) How effective is HADES across various VRPs compared with baselines (Table 1)? (ii) Why do *anisometric* positional encodings (IPE and CPE) outperform index-based alternatives (Tables 2 and 3)? (iii) Why does a *hierarchical* design matter (Table 5)? We also qualitatively analyze the effect of our anisometric positional encodings design by the visualization.

**Main Results** Table 1 summarizes the objectives and gaps (relative to HGS for CVRP and PyVRP-HGS for VRPTW/PCVRP) under identical time budgets. HADES outperforms prior methods in most settings and remains competitive elsewhere. At the largest scale ($N{=}2000$), HADES improves absolute gaps over strong OR baselines by about 2.45% on CVRP, 3.84% on VRPTW, and

Table 1: Results across instance sizes for CVRP, VRPTW, and PCVRP. Obj: objective, Gap: gap, Time: runtime (minutes). ↓ indicates lower is better. N/A indicates not for all instances; feasible solutions were found.

| Methods | CVRP 500 | | | CVRP 1000 | | | CVRP 2000 | | |
|---|---|---|---|---|---|---|---|---|---|
| | Obj.↓ | Gap↓ | Time↓ | Obj.↓ | Gap↓ | Time↓ | Obj.↓ | Gap↓ | Time↓ |
| HGS | 36.66 | - | 60s | 41.51 | - | 121s | 57.38 | - | 241s |
| LKH3 | 37.25 | 1.66% | 174s | 42.16 | 1.61% | 408s | 58.12 | 1.35% | 1448s |
| SISRs | 36.65 | 0.01% | 60s | 41.14 | -0.83% | 120s | 56.04 | -2.27% | 240s |
| NVIDIA cuOpt | 37.38 | 1.99% | 60s | 42.71 | 2.92% | 121s | 59.22 | 3.26% | 241s |
| BQ (BS64) | 37.51 | 2.32% | 23s | 43.32 | 4.36% | 164s | N/A | N/A | N/A |
| LEHD (RRC) | 37.04 | 1.04% | 60s | 42.47 | 2.31% | 121s | 60.11 | 4.76% | 246s |
| UDC | 37.63 | 2.69% | 60s | 42.65 | 3.68% | 121s | N/A | N/A | N/A |
| NCO-LLM | 36.93 | 0.74% | 60s | 41.96 | 1.08% | 121s | 59.43 | 3.57% | 246s |
| NDS | 36.57 | -0.20% | 60s | 41.11 | -0.90% | 120s | 56.00 | -2.34% | 240s |
| **HADES** | **36.54** | **-0.30%** | 60s | **41.05** | **-1.11%** | 120s | **55.98** | **-2.45%** | 240s |

| Methods | VRPTW 500 | | | VRPTW 1000 | | | VRPTW 2000 | | |
|---|---|---|---|---|---|---|---|---|---|
| | Obj.↓ | Gap↓ | Time↓ | Obj.↓ | Gap↓ | Time↓ | Obj.↓ | Gap↓ | Time↓ |
| PyVRP-HGS | 49.01 | - | 60s | 90.35 | - | 120s | 173.46 | - | 240s |
| SISRs | 48.09 | -1.87% | 60s | 87.68 | -2.98% | 120s | 167.49 | -3.49% | 240s |
| NVIDIA cuOpt | 49.30 | 2.60% | 61s | 90.31 | 0.03% | 121s | N/A | N/A | N/A |
| NDS | 47.94 | -2.17% | 60s | 87.54 | -3.14% | 120s | 167.48 | -3.50% | 240s |
| **HADES** | **47.90** | **-2.27%** | 60s | **87.45** | **-3.21%** | 120s | **166.80** | **-3.84%** | 240s |

| Methods | PCVRP 500 | | | PCVRP 1000 | | | PCVRP 2000 | | |
|---|---|---|---|---|---|---|---|---|---|
| | Obj.↓ | Gap↓ | Time↓ | Obj.↓ | Gap↓ | Time↓ | Obj.↓ | Gap↓ | Time↓ |
| PyVRP-HGS | 44.97 | - | 60s | 84.91 | - | 120s | 165.56 | - | 240s |
| SISRs | 43.22 | -3.90% | 60s | 81.12 | -4.55% | 120s | 158.17 | -4.54% | 240s |
| NVIDIA cuOpt | 43.34 | -3.71% | 60s | 81.89 | -3.74% | 121s | 160.33 | -3.37% | 241s |
| NDS | 43.12 | **-4.12%** | 60s | 80.99 | -4.71% | 120s | 158.09 | -4.60% | 240s |
| **HADES** | 43.13 | -4.09% | 60s | **80.77** | **-4.87%** | 120s | **157.37** | **-4.94%** | 240s |

Table 2: Ablation study results on CVRP 500 with different in-route PE methods employing various positional encoding algorithms.

Table 3: Generalization study on CVRP 500 with different node distribution and capacity distribution.

| Methods | Obj.↓ | Gap↓ |
|---|---|---|
| APE (Gehring et al., 2017) | 36.58 | -0.23% |
| RPE (Vaswani et al., 2017) | 36.57 | -0.24% |
| SIN (Shaw et al., 2018) | 36.55 | -0.28% |
| CPE (Ma et al., 2021) | 36.55 | -0.28% |
| RoPE (Su et al., 2024) | 36.56 | -0.26% |
| **HADES** | **36.54** | **-0.30%** |

| Methods | Low Capacity | | Cluster Distribution | |
|---|---|---|---|---|
| | Obj.↓ | Gap↓ | Obj.↓ | Gap↓ |
| HGS | 91.73 | - | 44.53 | - |
| SISRs | 91.34 | -0.38% | 44.31 | -0.49% |
| NDS | 91.15 | -0.59% | 44.29 | -0.54% |
| SIN | 91.08 | -0.70% | 44.25 | -0.63% |
| **HADES** | **91.04** | **-0.76%** | **44.22** | **-0.70%** |

4.94% on PCVRP, and it further surpasses NDS, which constructs the best known solutions. Although these benchmarks are already competitive, making additional progress increasingly difficult, our method still yields consistent, measurable gains. We attribute this to the hierarchical, anisometric positional encodings, which provide stronger global-local signals for deconstruction and translate into more effective deconstruction and reconstruction search.

**Effect of Positional Encoding** We compare five widely used positional encodings under identical training and testing, as shown in Table 2. APE (Gehring et al., 2017) directly assigns each node its absolute tour index; RPE (Vaswani et al., 2017) uses relative index offsets; SIN (Shaw et al., 2018) is the standard sinusoidal index mapping popularized in Transformers; CPE denotes the positional scheme introduced by Ma et al. (2021), which uses a cyclic input sequence to assign the position

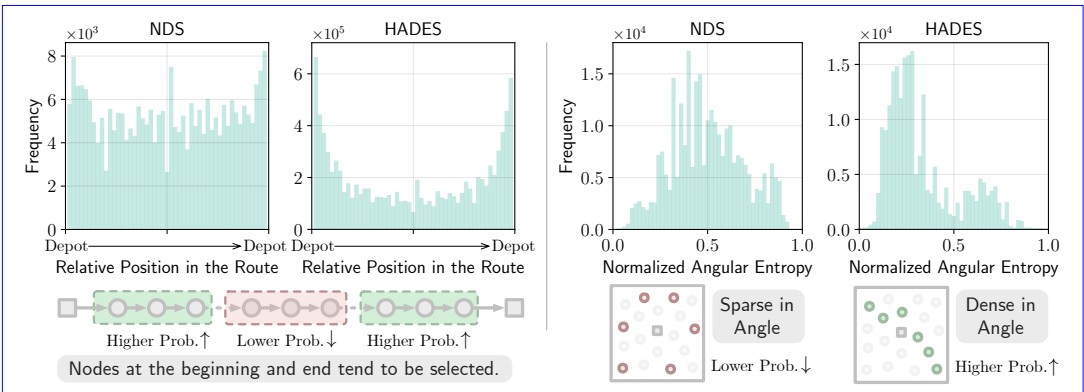

Figure 5: Geometric analysis on CVRP 500. Left: histogram of relative in-route positions of removed nodes. Right: distribution of normalized angular entropy per iteration.

encoding; and RoPE (Su et al., 2024) encodes positions by rotating queries and keys with a position-dependent phase so attention scores depend on relative offsets.

Our anisometric, distance-aware IPE attains the lowest objective and gap, indicating that grounding position in cumulative travel distance, normalized to a single period to respect route circularity, yields the most informative signal. Notably, APE and RPE underperform the encoder *without* any IPE, suggesting that naively injecting indices can be harmful: absolute scales drift with route length (e.g., length 10 vs. 50), creating distribution mismatch and forcing the network to learn instance- and route-specific rescalings. SIN and CPE are competitive but consistently worse than HADES, supporting the hypothesis that isometric, index-based encodings (or encodings not tied to actual traveled distance) are suboptimal for anisometric routes. RoPE also lags behind; plausible reasons include a mismatch between its intended use (multiplicative rotations inside attention) and our additive feature injection, sensitivity to variable tour lengths that induces frequency aliasing in index space, and the lack of explicit head-tail alignment unless carefully normalized. Overall, the results validate the design choice of a distance-aware, circularly consistent positional signal.

**Geometric Analysis** To enhance the interpretability of the learned deconstruction strategy, we analyze the geometric characteristics of the removed nodes during evaluation. We first examine their in-route positions. For each removed customer $v_i$ on a route $r = (v_1, \dots, v_L)$, we compute the normalized cumulative distance $\text{Pos}(v_i, r) = d_i/d_L$. Aggregating over 10M removals on CVRP 500, we observe that HADES preferentially removes customers near the beginning and end of routes as shown in Fig. 5. These boundary regions are where cross-route interactions and inefficient detours are more likely, focusing on deconstruction there results in more effective improvement.

We further study the angular distribution of removed nodes. For each iteration, we compute the angular histogram $p_\ell$ of all removed nodes and measure its entropy $H = -\sum_\ell p_\ell \log p_\ell$, reporting the normalized value $\tilde{H} = H/\log B$, where $B = 36$ is the number of angular bins. A lower $\tilde{H}$ indicates that removals concentrate within specific angular sectors. Using 200K evaluated solutions, we find that HADES exhibits substantially lower normalized angular entropy than NDS as shown in Fig. 5, suggesting that the policy tends to focus on a small number of coherent geometric regions rather than spreading removals uniformly across routes. This targeted, region-focused deconstruction effectively enlarges the local search neighborhood and contributes to greater improvements. For more geometric analysis on the other problems, please refer to Section B.6.

**Generalization** We test the generalization ability of the learned policy by transferring it beyond its training distribution on CVRP-500, as shown in Table 3. A model trained on a medium-capacity, uniform-location distribution is evaluated on (i) low-capacity instances and (ii) clustered customer layouts. HADES achieves the best objective and gap in both settings, compared with the state-of-the-art baseline, indicating robust generalization over changes in both capacity and spatial distribution.

In the low-capacity setting, a solution requires more routes, which intensifies inter-route interactions and creates dense, overlapping sectors (multiple tours may occupy a narrow angular span). The cross-route positional encoding supplies a route-level phase that helps the policy reason over this global layout, prioritize conflicting sectors, and select more effective removals. The comparison with NDS supports this claim with 91.04 vs. 91.15 in objective under low capacity.

Table 4: Results across instance sizes for PDTSP. Obj: objective, Gap: gap to LKH, Time: runtime.

| Methods | PDTSP-21 | | | PDTSP-51 | | | PDTSP-101 | | |
|---|---|---|---|---|---|---|---|---|---|
| | Obj.↓ | Gap↓ | Time↓ | Obj.↓ | Gap↓ | Time↓ | Obj.↓ | Gap↓ | Time↓ |
| LKH (5k) | 4.563 | 0.00% | 3m | 6.866 | 0.06% | 10m | 9.443 | 0.16% | 49m |
| LKH (10k) | 4.563 | 0.00% | 5m | 6.862 | 0.00% | 19m | 9.428 | 0.00% | 98m |
| Heter-AM (gr.) | 4.655 | 2.02% | 0s | 7.333 | 6.86% | 1s | 10.348 | 9.76% | 2s |
| Heter-AM (5k) | 4.578 | 0.33% | 33s | 7.108 | 3.58% | 1.5m | 10.051 | 6.61% | 5m |
| DACT (3k) | 4.564 | 0.03% | 1m | 7.057 | 2.83% | 1.5m | 10.195 | 8.13% | 2.5m |
| N2S (3k) | 4.565 | 0.05% | 1m | 7.027 | 2.40% | 1.5m | 9.846 | 4.44% | 3m |
| **HA-N2S** (3k) | **4.564** | **0.03**% | 1m | **6.997** | **1.96%** | 1.5m | **9.701** | **2.89%** | 3m |

Under clustered node distributions, intra-route distances become highly nonuniform as customers concentrate in localized regions. Replacing our distance-based IPE with the index-based sinusoidal encoding (SIN) degrades performance, indicating that equal-spacing assumptions are brittle when local spacing varies markedly. Grounding position in cumulative travel distance and enforcing circular phase alignment, combined with cross-route phases, preserves the underlying geometry and yields stronger out-of-distribution behavior.

**Model-agnostic Generalization**   To further examine the generalization capacity of our hierarchical anisometric encodings beyond the improvement framework used in HADES, we integrate them into N2S, which is a strong neural neighborhood search method for pickup-and-delivery problems (PDPs) (Ma et al., 2022). The resulting variant, denoted HA-N2S, replaces the original positional components of N2S with our hierarchical encoding. Following the experimental setup of the original N2S paper, we evaluate HA-N2S on PDTSP instances of sizes 21, 51, and 101. As summarized in Table 4, HA-N2S consistently outperforms the vanilla N2S across all instance sizes without additional computational overhead. These results indicate that both distance-aware anisometry and hierarchical cross-route structure are essential for capturing the geometric organization of PDP solutions. Moreover, the gains observed on PDTSP demonstrate that our encoding scheme serves as a plug-and-play enhancement for diverse NCO policies, reinforcing its model-agnostic applicability. For more experimental details, please refer to Section B.5.

**Ablation study**   Table 5 quantifies the contribution of each of our hierarchical encoding components on CVRP 500. Removing either component degrades performance. Thus, both in-route and cross-route signals matter, with the cross-route encoding providing the larger marginal gain in this setting. Since these instances are already near-optimal, even these small percentages represent meaningful improvements, highlighting that the hierarchical positional design supplies useful guidance for deconstruction.

Table 5: Ablation study on CVRP 500.

| Methods | Obj.↓ | Gap↓ |
|---|---|---|
| **HADES** | **36.54** | **-0.30%** |
| w/o IPE | 36.56 | -0.27% |
| w/o CPE | 36.57 | -0.23% |

## 6   CONCLUSION

We presented HADES, a neural improvement framework for VRPs that aligns representation with problem geometry via two anisometric positional encodings: a distance-indexed in-route signal that respects tour circularity and a depot-anchored cross-route phase that captures global route arrangement. Coupled with a lightweight encoder, a history-aware pointer decoder, and batched ASA search, this hierarchical design improves solution quality across various VRPs under equal time budgets, outperforming strong OR heuristics and prior neural methods. Analyses and ablations show that both components matter and that grounding position in cumulative distance and relative angle yields robust generalization under distribution shifts. The takeaway is simple: learning strong order representations for the solution allows for more effective search in the solution space itself.

A current limitation is that the improvement phase runs on a single CPU core; parallelizing deconstruction, repair, and acceptance should increase throughput and tighten quality.

REPRODUCIBILITY STATEMENT

We have made every effort to ensure the reproducibility of our results. Detailed descriptions of model architectures, training procedures, and experimental setups are provided in both the main paper and the appendix to guarantee reproducibility. Finally, all code to reproduce the experiments will be released open-source upon acceptance.

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

# A  DETAILED DEFINITIONS

## A.1  VEHICLE ROUTING PROBLEMS

We consider a complete directed graph $G = (V, E)$ with $V = \{0, 1, \ldots, N\}$, where $x_0$ is the depot and $x_i$ for $i \geq 1$ are customers. Let $\mathbf{x}_i \in \mathbb{R}^2$ denote the coordinates of node $x_i$ and define travel cost $c_{ij} = \|\mathbf{x}_i - \mathbf{x}_j\|_2$ for $(i, j) \in E$. A fleet $K = \{1, \ldots, M\}$ of identical vehicles of capacity $C$ is available. For convenience, we use binary routing variables $x_{ij}^k \in \{0, 1\}$ indicating whether vehicle $k$ traverses arc $(i, j)$ and service variables $y_i^k \in \{0, 1\}$ indicating whether vehicle $k$ serves customer $i$. We also write $y_i = \sum_{k \in K} y_i^k$.

**Capacitated VRP (CVRP).**  Each customer $i \in \{1, \ldots, N\}$ has demand $d_i \geq 0$. The objective is to minimize total travel distance:

$$\min_{x,y} \quad \sum_{k \in K} \sum_{i \in V} \sum_{j \in V} c_{ij}\, x_{ij}^k. \tag{4}$$

Subject to service, capacity, flow, and depot constraints:

$$\sum_{k \in K} y_i^k = 1 \qquad\qquad \forall i \in \{1, \ldots, N\}, \tag{CVRP-1a}$$

$$\sum_{i \in \{1,\ldots,N\}} \sum_{j \in V} d_i\, x_{ij}^k \leq C \qquad\qquad \forall k \in K, \tag{CVRP-1b}$$

$$\sum_{j \in V} x_{ij}^k = y_i^k, \quad \sum_{j \in V} x_{ji}^k = y_i^k \qquad \forall i \in \{1, \ldots, N\},\ \forall k \in K, \tag{CVRP-1c}$$

$$\sum_{j \in V} x_{0j}^k = 1, \quad \sum_{i \in V} x_{i0}^k = 1 \qquad \forall k \in K, \tag{CVRP-1d}$$

$$x_{ij}^k \in \{0, 1\}, \quad y_i^k \in \{0, 1\} \qquad \forall i, j \in V,\ \forall k \in K. \tag{CVRP-1e}$$

Constraints (CVRP-1a) ensure each customer is served exactly once. Constraints (CVRP-1b) enforce per-vehicle capacity. Constraints (CVRP-1c) link service and flow to prevent disjoint subtours around customers. Constraints (CVRP-1d) ensure each vehicle departs from and returns to the depot once. Standard subtour-elimination constraints (e.g., MTZ or flow-based) can be added if needed for exact optimization but are omitted here for brevity.

**VRP with Time Windows (VRPTW).**  In addition to CVRP data, each customer $i$ has a service duration $s_i \geq 0$ and an admissible time window $[a_i, b_i]$ within which service must start. Early arrival is allowed (vehicles may wait), tardiness is not. Let travel time $\tau_{ij} = c_{ij}$ and introduce service start times $t_i \in \mathbb{R}_{\geq 0}$ for $i \in V$ (with $t_0$ the depot time). The objective is to minimize total travel time:

$$\min_{x,y,t} \quad \sum_{k \in K} \sum_{i \in V} \sum_{j \in V} \tau_{ij}\, x_{ij}^k. \tag{5}$$

Subject to CVRP constraints (CVRP-1b)–(CVRP-1e), a relaxed service constraint, and time-feasibility:

$$\sum_{k \in K} y_i^k = 1 \qquad\qquad \forall i \in \{1, \ldots, N\}, \tag{VRPTW-2a}$$

$$a_i \leq t_i \leq b_i \qquad\qquad \forall i \in \{1, \ldots, N\}, \tag{VRPTW-2b}$$

$$t_j \geq t_i + s_i + \tau_{ij} - M_{\text{tw}}\Big(1 - \sum_{k \in K} x_{ij}^k\Big) \qquad \forall i, j \in V, \tag{VRPTW-2c}$$

where $M_{\text{tw}}$ is a sufficiently large constant. Constraint (VRPTW-2b) enforces time windows; (VRPTW-2c) imposes temporal precedence whenever arc $(i, j)$ is used. Early arrival is implicitly modeled by the lower bound $t_i \geq a_i$, with waiting if $t_i$ is set earlier than feasible departure due to successors.

**Prize-Collecting VRP (PCVRP).** Customers carry prizes $p_i \geq 0$ and need not all be served. Let $y_i \in \{0, 1\}$ indicate whether customer $i$ is visited by any vehicle. The objective trades off travel cost and collected prizes:

$$\min_{x,y} \quad \sum_{k \in K} \sum_{i \in V} \sum_{j \in V} c_{ij} \, x_{ij}^k \;-\; \sum_{i \in \{1,...,N\}} p_i \, y_i. \tag{6}$$

Subject to capacity, flow, and depot constraints as in CVRP, plus visit linking and at-most-once service:

$$\sum_{k \in K} \sum_{j \in V} x_{ij}^k = y_i, \quad \sum_{k \in K} \sum_{j \in V} x_{ji}^k = y_i \qquad \forall i \in \{1, \ldots, N\}, \tag{PCVRP-3a}$$

$$y_i \in \{0, 1\}, \quad x_{ij}^k \in \{0, 1\} \qquad \forall i, j \in V, \; \forall k \in K. \tag{PCVRP-3b}$$

Constraints (PCVRP-3a) tie $y_i$ to the vehicle flows so that a customer is counted as served if and only if exactly one visit occurs. Capacity and depot constraints bound feasible prize-collecting tours.

These formulations are consistent with the main-text notation: costs derive from Euclidean distances $c_{ij} = \|\mathbf{x}_i - \mathbf{x}_j\|_2$, vehicles have capacity $C$, time windows $[a_i, b_i]$ with service $s_i$ enforce feasibility for VRPTW, and prizes $p_i$ define the trade-off in PCVRP. In our experiments, travel time equals distance, and vehicles are identical; implementation details (e.g., exact subtour elimination, big-$M$ values, and generator-specific normalization of demands and capacities) are provided in the appendix section on data and generator settings.

### A.2 Markov Decision Process Formulation for Deconstruction Method

We model deconstruction as a Markov Decision Process (MDP) in which a policy iteratively removes customers from a feasible solution to reduce total travel cost.

**State space**  Let $\mathbf{s}_t$ denote the current feasible solution at step $t$ for instance $l$. We encode it as

$$s_t \;=\; \Psi(l, \mathbf{s}_t) \;=\; \big\{\, x_0, \ldots, x_N; \; d_1, \ldots, d_N; \; p_{t1}, \ldots, p_{tN} \,\big\},$$

where $x_i$ is the customer node (with coordinates $\mathbf{x}_i \in \mathbb{R}^2$ and $x_0$ the depot), $d_i$ is demand, and $p_{ti}$ aggregates solution-aware positional features used by our encoder (e.g., in-route IPE for node $i$ and the broadcast cross-route CPE for its route at time $t$). The state implicitly includes the current route partition and the mask of already-removed or infeasible customers.

**Action space**  At step $t$ the action selects one customer for removal:

$$x_t \in \{1, \ldots, N\},$$

where $x_t$ denotes the index of the customer node chosen at step $t$ (the depot 0 is never selectable). Across a removal horizon of $M$ steps, the action sequence is $x_{1:t-1}$ for history and $x_t$ for the current choice.

**Transition**  Given $x_t$, the next solution is obtained by removing the selected customer from $\mathbf{s}_t$ and re-marking feasibility:

$$\mathbf{s}_{t+1} \;=\; \mathcal{T}(\mathbf{s}_t, x_t), \qquad s_{t+1} = \Psi(l, \mathbf{s}_{t+1}).$$

The transition is deterministic; invalid actions are masked in the policy.

**Reward**  Let $\mathrm{cost}(\mathbf{s})$ be the objective (distance or time). With incumbent best $\mathbf{s}_t^\star = \arg\min\{\mathrm{cost}(\mathbf{s}_0), \ldots, \mathrm{cost}(\mathbf{s}_t)\}$, we use an improvement reward

$$r_t \;=\; \mathrm{cost}(\mathbf{s}_t^\star) \;-\; \min\big\{\, \mathrm{cost}(\mathbf{s}_{t+1}), \, \mathrm{cost}(\mathbf{s}_t^\star) \,\big\},$$

which is strictly positive iff a new incumbent is found. This shape makes the undiscounted return equal to the total cost reduction over the initial solution.

**Policy**  The stochastic policy is conditioned on the instance, the current solution, a rollout seed $v \in \{0, 1\}^{d_v}$, and the action history:

$$\pi_\theta\big(x_t \,\big|\, l, \, \mathbf{s}_t, \, v, \, x_{1:t-1}\big),$$

where $\theta$ are network parameters. Actions are sampled during training and selected greedily or stochastically at test time; masking enforces feasibility.

**MDP summary** The deconstruction MDP is $(\mathcal{S}, \mathcal{A}, \mathcal{T}, r, \pi_\theta)$ with

$$\mathcal{S} = \{\Psi(l, \mathbf{s})\}, \quad \mathcal{A} = \{1, \ldots, N\}, \quad \mathbf{s}_{t+1} = \mathcal{T}(\mathbf{s}_t, x_t), \quad r_t = \text{cost}(\mathbf{s}_t^\star) - \min\{\text{cost}(\mathbf{s}_{t+1}), \text{cost}(\mathbf{s}_t^\star)\}.$$

This formulation matches the main text: the state encodes instance and solution features (including IPE/CPE), the action removes a single customer $x_t$, transitions are deterministic with masking, and the reward directly reflects improvement in routing cost.

### A.3    ANISOMETRIC

In this work, we use the term anisometric to refer to sequential structures in which adjacent elements exhibit non-uniform geometric separations rather than the equal-step spacing implicitly assumed by index-based isometric positional encodings. Consider a route $r = (v_1, \ldots, v_L)$ with node coordinates $x_{v_i} \in \mathbb{R}^2$. The Euclidean distances between consecutive nodes, $\|x_{v_{i+1}} - x_{v_i}\|_2$, generally vary with $i$, and the corresponding cumulative-distance mapping $d_i = \sum_{j=2}^{i} \|x_{v_j} - x_{v_{j-1}}\|_2$ becomes a nonlinear function of the discrete index $i$. As a result, increments in the index do not translate to uniform increments in physical space. We refer to such structures as anisometric to emphasize that VRP routes possess heterogeneous spatial separations that must be represented explicitly when designing positional encodings.

## B    ADDITIONAL EXPERIMENTAL DETAILS

### B.1    DATA GENERATION

**CVRP instances.** Node coordinates $\{\mathbf{x}_i\}_{i=0}^{N}$ are sampled either uniformly in the unit square (default) or from a clustered generator (as specified in the main text). Customer demands are discrete, $d_i \sim \text{Unif}\{1, \ldots, 9\}$, and vehicles are identical with capacity $C$. Following common NCO practice (Kool et al., 2019; Kwon et al., 2020), we set a size-dependent capacity

$$C = \begin{cases} 30 + \left\lfloor \frac{1000}{5} + \frac{n-1000}{33.3} \right\rfloor, & n > 1000, \\ 30 + \left\lfloor \frac{n}{5} \right\rfloor, & 20 < n \leq 1000, \\ 30, & \text{otherwise,} \end{cases}$$

which yields comparable route loads across sizes. Travel cost is Euclidean distance $c_{ij} = \|\mathbf{x}_i - \mathbf{x}_j\|_2$ and speed is set to 1 so that time equals distance.

**PCVRP instances.** We reuse the CVRP location/demand generators and attach a prize to each customer that scales with demand:

$$p_i = \kappa \xi_i d_i, \qquad \xi_i \sim \text{Unif}[0.8, 1.2],$$

with a global scale $\kappa$ chosen so that, under a simple greedy oracle, roughly half of the customers are economically attractive to visit (keeps difficulty balanced). The objective minimized at evaluation is $\sum_{k,i,j} c_{ij} x_{ij}^k - \sum_i p_i y_i$; all other modeling choices (speed, costs) match CVRP.

**VRPTW instances.** We reuse the CVRP location/demand generators and then synthesize time windows $[e_i, l_i]$ and service times $s_i$ for each customer $i$. Let $d_{ij} = \|\mathbf{x}_i - \mathbf{x}_j\|_2$ and assume unit speed so travel time equals distance. The procedure is:

1. Sample service times $s_i \sim \text{Unif}[0.15, 0.18]$.
2. Sample window lengths $t_i \sim \text{Unif}[0.18, 0.20]$.
3. Compute a distance scale to the nearest horizon via the farthest depot distance $d_i^{\max} = \max_{j \in \{0, \ldots, m-1\}} d_{ij}$ (for single-depot instances $d_i^{\max} = d_{i0}$).
4. Given a global horizon $t_{\max}$, set an upper bound on feasible starts $h_i = \frac{t_{\max} - s_i - t_i}{d_i^{\max}} - 1$.
5. Draw a start fraction $u_i \sim \text{Unif}[0, 1]$ and set the window start $e_i = \left(1 + (h_i - 1)u_i\right) d_i^{\max}$.
6. Set the window end $l_i = e_i + t_i$.

During decoding, infeasible moves are masked using the standard check that the expected service start at $j$, $\max(t_{\text{curr}} + d_{ij}, e_j)$, plus service $s_j$ does not exceed the deadline $l_j$. For closed tours we also enforce return feasibility:

$$\max(t_{\text{curr}} + d_{ij}, e_j) + s_j \leq l_j, \qquad \max(t_{\text{curr}} + d_{ij}, e_j) + s_j + d_{j0} \leq l_0.$$

As an alternative, one may adopt the Solomon procedure (Solomon, 1987) (e.g., Li et al., 2021; Zhou et al., 2024); our code provides both options.

## B.2 BASELINES

**HGS** (Vidal, 2022). Hybrid Genetic Search is a state-of-the-art metaheuristic for CVRP that combines population-based recombination with powerful local search (e.g., ejection chains, $k$-opt moves, route exchanges) and adaptive penalty management. We use the public implementation with settings recommended by the authors.

**SISRs** (Christiaens & Vanden Berghe, 2020). Slack Induction by String Removals is a general ruin-and-recreate framework. It removes contiguous strings of customers to induce temporal or capacity slack and then repairs the solution with tailored insertion heuristics. Parameters are adapted online to balance diversification and intensification.

**LKH3** (Helsgaun, 2017). The LKH heuristic extends the classic $k$-opt search with sophisticated candidate sets and powerful move pruning. Although historically focused on TSP, it includes support for several VRP variants and remains a strong baseline for large-scale routing.

**PyVRP** (Wouda et al., 2024). PyVRP is an open-source Python/C++ solver that implements a modern variant of HGS together with route-improvement neighborhoods and feasibility handling. It supports capacity, time windows, and many side constraints. We use version 0.9.0 and the authors' default configuration.

**NVIDIA cuOpt** (NVIDIA Corporation, 2025). cuOpt is a GPU-accelerated local-search engine for rich VRPs. It exploits massive parallelism to evaluate neighborhoods and apply route moves at scale, yielding high-quality solutions under tight time budgets. We follow the vendor's recommended settings for our variants.

**BQ** (Drakulic et al., 2024). BQ is a re-encoding construction policy: after each partial decision the model re-encodes the remaining subproblem and continues decoding, improving distributional fit and enabling stronger lookahead than one-shot construction. It represents a strong neural baseline on CVRP. We test BQ with beam search with beam width of 64 (BS64).

**LEHD** (Luo et al., 2023). LEHD is a neural construction framework that augments attention decoding with heuristic-informed logit shaping and hierarchical decision rules. It improves stability and search efficiency, especially on large instances where pure neural decoders struggle. We test LEHD with Random Route Reconstruction (RRC).

**UDC** (Zheng et al., 2024). Universal Divide-and-Conquer decomposes large routing instances into subregions, solves them with a learned policy (possibly in parallel), and then merges and polishes the partial tours. The approach targets scalability while maintaining competitive quality.

**NCO-LLM** (Tran et al., 2025). NCO-LLM is a neural construction framework based on LEHD Luo et al. (2023) which employs LLM-designed (Liu et al., 2024; Ye et al., 2024a) heuristics for output logit reshaping, enhancing large-scale generalization and overall performance. We test NCO-LLM with the same RRC as LEHD.

**NDS** (Hottung et al., 2025b). Neural Deconstruction Search is a learning-to-improve method that trains a policy to select customers for removal; reconstruction and acceptance are managed by an augmented simulated annealing loop. Execution is batched on GPU, making it a strong prior art for learned improvement.

Table 6 lists the used assets and their licenses. Our code will be licensed under the MIT License.

| Asset | License & Usage |
|---|---|
| **Classical OR Solvers** | |
| HGS (Vidal, 2022) | MIT License |
| SISRs (Christiaens & Vanden Berghe, 2020) | MIT License |
| LKH3 (Helsgaun, 2017) | Academic, non-commercial use; |
| PyVRP (Wouda et al., 2024) | MIT License |
| NVIDIA cuOpt | Apache-2.0 |
| **Neural Baselines** | |
| SGBS-EAS (Choo et al., 2022) | MIT License |
| LEHD (Luo et al., 2023) | Non-commercial research use |
| UDC (Zheng et al., 2024) | Repository code available, no claim; |
| NCO-LLM (Tran et al., 2025) | MIT License |
| NDS (Hottung et al., 2025b) | Repository code available, no claim; |

Table 6: Baselines and license information used in our experiments.

## B.3 TRAINING CONFIGURATION

We train HADES on CVRP using REINFORCE with softmax sampling, automatic mixed precision, and gradient accumulation. At each training step, we sample binary latent vectors $z \in \{0, 1\}^{10}$ and execute rollouts over partially destroyed solutions. The policy receives an advantage signal formed by subtracting the per-instance mean across rollouts; only the best rollout per instance contributes to the policy gradient. We remove 15 nodes per rollout and apply a destroy/repair operator.

The HADES encoder augments node embeddings with tour-aware signals (tour id/position, previous/next neighbors, cumulative tour distance at arrival, and depot-relative angle). The encoder uses two self-attention layers, a message-passing layer over tour neighbors, and two additional self-attention layers. The decoder employs a GRU context, multi-head attention, a small polynomial head that conditions on $z$, and logit clipping before softmax. Optimization uses Adam with Multi-StepLR and a 5,000-step linear warmup; gradients are accumulated to align optimizer updates with the inner search horizon.

## B.4 TESTING CONFIGURATION

For completeness, we summarize the evaluation protocol used in all experiments. Each test instance is evaluated once under a fixed wall-clock budget, following the standard setup of recent neural improvement methods such as NDS (Hottung et al., 2025b). For completeness, we report the variance across runs of our method. For each setting, we train three independent models under identical configurations and evaluate all of them on the same test sets used in the main paper. Table 8 reports the mean and standard deviation of the objective values across these three runs. Baseline methods do not report variance or multiple-run statistics in their original papers. Since all approaches are evaluated on the same standardized test datasets, we believe this comparison remains fair and consistent with current evaluation practices in neural improvement and VRP research.

Our method is executed on a single CPU core with a single Nvidia A100 GPU and is given the same wall-clock limits (60, 120, or 240 seconds depending on problem size) as baseline methods. The parameters governing augmentations, rollout counts, and reconstruction attempts are fixed throughout and described in the main text. This configuration ensures fair comparison across solvers while maintaining consistency with established test-time evaluation protocols in the literature.

## B.5 HA-N2S CONFIGURATION

For the HA-N2S experiment, we follow the original N2S setup (Ma et al., 2022) in all training, model, and inference configurations, replacing only the positional encoding with our proposed hierarchical anisometric embedding.

**Model architecture.** The encoder and decoder structures, including the Synthesis Attention module and the two PDP-specific decoders, remain unchanged. The only modification is substituting the

| Hyperparameter | Value |
|---|---|
| **Optimization** | |
| Optimizer | Adam |
| Initial learning rate | $10^{-4}$ |
| Weight decay | $10^{-6}$ |
| LR scheduler | MultiStepLR (milestones: 60, 85; $\gamma = 0.1$) |
| Warmup | Linear, 5,000 steps (start factor 0.01) |
| Mixed precision | Yes (AMP) |
| Gradient accumulation steps | 100 |
| Optimizer step interval | Every 100 inner updates |
| **Neural Architecture** | |
| Embedding dimension | 128 |
| Number of attention heads | 8 |
| QKV dimension | 16 |
| Message passing layers | 1 |
| Encoder layers | 2 |
| Feedforward dimension | 512 |
| Decoder | GRUCell(128→128) + attention |
| Polynomial head dim | 256 |
| Logit clipping | 10 |
| Latent vector ($z$) dimension | 10 |
| Inference rule (training) | Softmax sampling |
| **Training Dynamics** | |
| Total training epochs | 2000 |
| Iterations per epoch | 150,000 |
| Batch size | 64 |
| Rollout size (per instance) | 128 |
| Skipped iterations (search warmup) | 10 |
| Nodes removed per rollout | 15 |
| Destroy/repair recreate parameter | 5 |
| Reward type | Improvement (absolute) |
| **Evaluation** | |
| Validation episodes | 1,000 |
| Iterations per episode | 50 |
| Validation batch size | 50 |
| Rollouts per instance (validation) | 200 |
| Augmented rollouts | $8\times$ (8-fold geometric augmentation) |

Table 7: Training and evaluation hyperparameters for HADES.

| Problem | 500 | 1000 | 2000 |
|---------|-----|------|------|
| CVRP | $36.54 \pm 0.01$ | $41.05 \pm 0.03$ | $55.98 \pm 0.11$ |
| VRPTW | $47.90 \pm 0.03$ | $87.45 \pm 0.08$ | $166.80 \pm 0.23$ |
| PCVRP | $43.13 \pm 0.01$ | $80.77 \pm 0.06$ | $157.37 \pm 0.21$ |

Table 8: Objective mean and standard deviation of our method across three independent training runs.

cyclic positional encoding with our hierarchical anisometric embedding, while keeping the embedding dimensionality and all other components identical to the original implementation.

**Training configuration.** We adopt the same PPO-based training procedure used in N2S, including curriculum learning, batch size, number of epochs, training horizon, and optimization hyperparameters. Models are trained separately for $|V| = 21, 51, 101$ using the same training schedule as in Ma et al. (2022), without introducing any additional tuning beyond replacing the positional input.

**Inference configuration.** During inference, we use the same augmentation-based search scheme and the same number of search steps (1k, 2k, 3k) as reported in the original N2S evaluation. All feasibility constraints, masking rules, and reinsertion operations are kept fully consistent with the N2S framework.

This setup ensures that the HA-N2S results isolate the effect of the hierarchical anisometric embedding while preserving full comparability with the original N2S configuration.

### B.6 Additional Discussion

**NCO-LLM** Recent progress in LLM-assisted routing has introduced baselines such as NCO-LLM, ReEvo-ACO, and LLM-driven large-neighborhood heuristics, which integrate large language models to design or refine heuristic components within established neural or metaheuristic frameworks. NCO-LLM, in particular, augments the LEHD construction policy by using LLM-generated rules to reshape output logits, thereby improving its large-scale generalization and decoding stability. While these approaches demonstrate that LLMs can enhance heuristic quality, they remain fundamentally constructive or heuristic-guided methods. In contrast, our framework is an improvement-based approach that operates directly in the solution space and incorporates anisometric hierarchical positional encodings tailored to VRP structure. The empirical results indicate that our method achieves stronger performance under identical runtime limits, suggesting that explicit modeling of route geometry and solution-aware relational structure can provide benefits that complement, and in several cases surpass, the heuristic guidance enabled by LLMs.

$r^*$ **Selection** The cross-route positional encoding uses a reference route $r^*$ solely to establish a phase origin from which relative angular offsets between routes are computed. This choice carries no semantic meaning because the encoding depends only on angular differences rather than absolute orientation. This design principle mirrors that of modern positional encoding schemes in sequence models such as GPT, Qwen, and DeepSeek, where absolute positional anchors are arbitrary and the model operates entirely on relative structure. Using a random route as $r^*$ prevents the introduction of an artificial geometric bias that would arise from deterministic selection rules. Since all information used by the model is expressed through relative angular relationships, the encoder's behavior is insensitive to which route is chosen as $r^*$.

**Additional Geometric Analysis** We provide additional geometric analysis for VRPTW 500 and PCVRP 500 to complement the CVRP results presented in the main paper. As shown in Fig. 6, the in-route positional distributions again reveal that our model concentrates removals near the beginning and end of routes, while NDS exhibits a more uniform pattern. This consistent boundary-focused behavior across problem variants supports the role of hierarchical anisometric encodings in capturing route-level structure that guides more effective deconstruction. Likewise, the normalized angular entropy demonstrates the same trend observed in CVRP: our method produces substantially lower entropy than NDS, indicating that removals are concentrated within a small set of coherent angular sectors rather than being dispersed across the full angular range. This focused angular behavior

appears across both VRPTW and PCVRP despite their additional problem-specific constraints, suggesting that the learned geometric patterns generalize beyond the CVRP setting and contribute to the stronger improvement performance observed on these tasks.

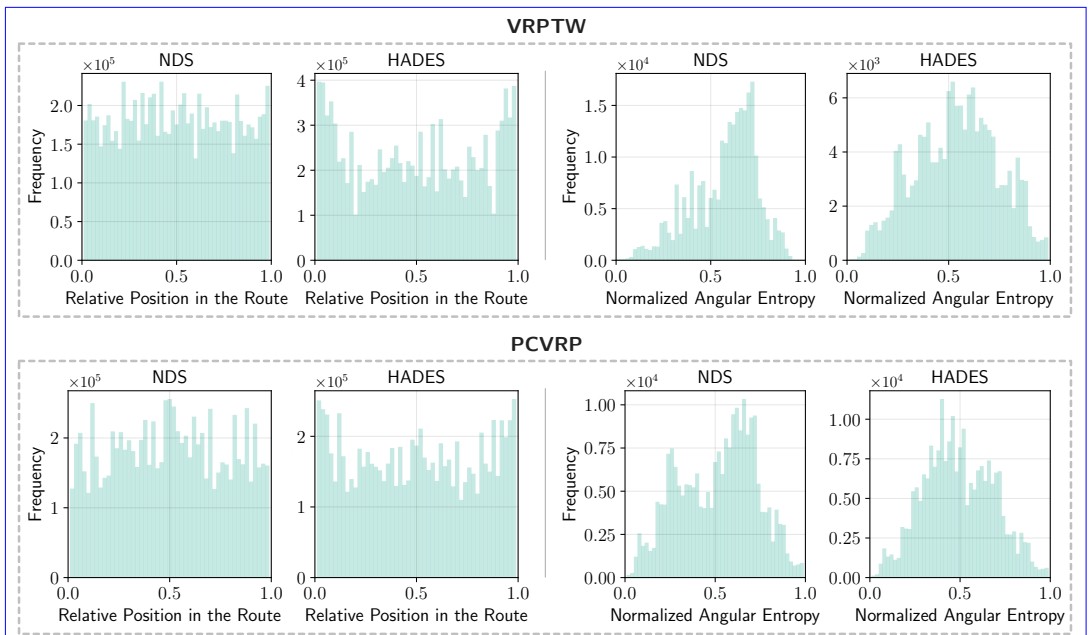

Figure 6: Additional geometric analysis on VRPTW 500 and PCVRP 500. For each problem, left: histogram of relative in-route positions of removed nodes; right: distribution of normalized angular entropy per iteration.

## B.7 EVALUATION ON CVRPLIB

To further assess the generalization capability of our method, we additionally evaluate the model on the X instances from CVRPLib (Lima et al., 2014), a widely adopted benchmark suite with heterogeneous spatial distributions, varied customer densities, diverse numbers of customers ($N \in [100, 1000]$), and overall topologies. We directly apply the checkpoints of NDS and HADES trained on $N = 500$, without any retraining or size-specific adaptation, and with a time limit of 60 seconds. As shown in Table 9, our model attains an average gap of $2.828\%$ across the full set, substantially improving over NDS, which yields an average gap of $4.110\%$ under the same evaluation protocol. These results indicate that the hierarchical anisometric encodings learned on synthetic CVRP effectively transfer to real benchmark distributions that differ in geometry, customer placement, and route topology. The performance gain observed here provides empirical evidence that our method generalizes robustly without per-size training to out-of-distribution settings, and complements the additional distribution-shift evaluations reported in the main paper.

## C USE OF LARGE LANGUAGE MODELS

Large language models (LLMs) were employed as general-purpose writing assistants. Their use was restricted to refining phrasing, improving clarity, and correcting grammar in draft versions of the manuscript. Research ideas, methodologies, analyses, results, and interpretations were conceived and validated solely by the authors. Furthermore, any text generated with the assistance of LLMs was thoroughly reviewed, edited, and integrated by the authors to ensure accuracy, correctness, and compliance with academic standards.

Table 9: Comparisons on the X set of CVRPLib: best-known solutions (BKS), NDS and HADES costs, and their percentage gaps. NDS shows clear improvements over NDS in out-of-distribution, real-world settings.

| Instance | BKS | NDS Obj. | NDS Gap% | HADES Obj. | HADES Gap% | Instance | BKS | NDS Obj. | NDS Gap% | HADES Obj. | HADES Gap% |
|---|---|---|---|---|---|---|---|---|---|---|---|
| X-n101-k25 | 27591 | 27696 | 0.381 | 27696 | 0.381 | X-n401-k29 | 66154 | 68272 | 3.202 | 67508 | 2.047 |
| X-n106-k14 | 26362 | 26711 | 1.324 | 26567 | 0.778 | X-n411-k19 | 19712 | 20837 | 5.707 | 20490 | 3.947 |
| X-n110-k13 | 14971 | 15165 | 1.296 | 15082 | 0.741 | X-n420-k130 | 107798 | 112070 | 3.963 | 110825 | 2.808 |
| X-n115-k10 | 12747 | 12765 | 0.141 | 12762 | 0.118 | X-n429-k61 | 65449 | 68765 | 5.067 | 67692 | 3.427 |
| X-n120-k6 | 13332 | 13595 | 1.973 | 13518 | 1.395 | X-n439-k37 | 36391 | 37377 | 2.709 | 37085 | 1.907 |
| X-n125-k30 | 55539 | 56943 | 2.528 | 56551 | 1.822 | X-n449-k29 | 55233 | 57363 | 3.856 | 56606 | 2.486 |
| X-n129-k18 | 28940 | 29326 | 1.334 | 29190 | 0.864 | X-n459-k26 | 24139 | 25361 | 5.062 | 24966 | 3.426 |
| X-n134-k13 | 10916 | 11110 | 1.777 | 11050 | 1.228 | X-n469-k138 | 221824 | 246070 | 10.930 | 238130 | 7.351 |
| X-n139-k10 | 13590 | 13699 | 0.802 | 13643 | 0.390 | X-n480-k70 | 89449 | 93513 | 4.543 | 92099 | 2.963 |
| X-n143-k7 | 15700 | 16160 | 2.930 | 16000 | 1.911 | X-n491-k59 | 66483 | 69124 | 3.972 | 68373 | 2.843 |
| X-n148-k46 | 43448 | 44351 | 2.078 | 44076 | 1.445 | X-n502-k39 | 69226 | 70762 | 2.219 | 70297 | 1.547 |
| X-n153-k22 | 21220 | 21609 | 1.833 | 21466 | 1.159 | X-n513-k21 | 24201 | 25313 | 4.595 | 24952 | 3.103 |
| X-n157-k13 | 16876 | 16957 | 0.480 | 16957 | 0.480 | X-n524-k153 | 154593 | 161430 | 4.423 | 159215 | 2.990 |
| X-n162-k11 | 14138 | 14262 | 0.877 | 14210 | 0.509 | X-n536-k96 | 94846 | 98716 | 4.080 | 97547 | 2.848 |
| X-n167-k10 | 20557 | 21012 | 2.213 | 20905 | 1.693 | X-n548-k50 | 86700 | 93650 | 8.016 | 91431 | 5.457 |
| X-n172-k51 | 45607 | 46201 | 1.302 | 46048 | 0.967 | X-n561-k42 | 42717 | 44239 | 3.563 | 43732 | 2.376 |
| X-n176-k26 | 47812 | 49478 | 3.484 | 48940 | 2.359 | X-n573-k30 | 50673 | 52296 | 3.203 | 51740 | 2.106 |
| X-n181-k23 | 25569 | 25966 | 1.553 | 25806 | 0.927 | X-n586-k159 | 190316 | 206204 | 8.348 | 201364 | 5.805 |
| X-n186-k15 | 24145 | 24364 | 0.907 | 24312 | 0.692 | X-n599-k92 | 108451 | 115105 | 6.135 | 113179 | 4.360 |
| X-n190-k8 | 16980 | 17536 | 3.274 | 17377 | 2.338 | X-n613-k62 | 59535 | 62191 | 4.461 | 61294 | 2.955 |
| X-n195-k51 | 44225 | 45025 | 1.809 | 44750 | 1.187 | X-n627-k43 | 62164 | 66173 | 6.449 | 65024 | 4.601 |
| X-n200-k36 | 58578 | 60847 | 3.873 | 60041 | 2.498 | X-n641-k35 | 63684 | 66650 | 4.657 | 65782 | 3.294 |
| X-n204-k19 | 19565 | 19924 | 1.835 | 19814 | 1.273 | X-n655-k131 | 106780 | 110878 | 3.838 | 109468 | 2.517 |
| X-n209-k16 | 30656 | 31059 | 1.315 | 30973 | 1.034 | X-n670-k130 | 146332 | 160574 | 9.733 | 156357 | 6.851 |
| X-n214-k11 | 10856 | 11236 | 3.500 | 11102 | 2.266 | X-n685-k75 | 68205 | 70980 | 4.069 | 70186 | 2.904 |
| X-n219-k73 | 117595 | 120258 | 2.265 | 119247 | 1.405 | X-n701-k44 | 81923 | 85467 | 4.326 | 84368 | 2.985 |
| X-n223-k34 | 40437 | 41043 | 1.499 | 40885 | 1.108 | X-n716-k35 | 43373 | 45545 | 5.008 | 44907 | 3.537 |
| X-n228-k23 | 25742 | 26685 | 3.663 | 26389 | 2.513 | X-n733-k159 | 136187 | 143437 | 5.324 | 141241 | 3.711 |
| X-n233-k16 | 19230 | 19726 | 2.579 | 19594 | 1.893 | X-n749-k98 | 77269 | 81463 | 5.428 | 80249 | 3.857 |
| X-n237-k14 | 27042 | 27971 | 3.435 | 27726 | 2.529 | X-n766-k71 | 114417 | 122799 | 7.326 | 120327 | 5.165 |
| X-n242-k48 | 82751 | 84082 | 1.608 | 83717 | 1.167 | X-n783-k48 | 72386 | 76158 | 5.211 | 75081 | 3.723 |
| X-n247-k50 | 37274 | 38867 | 4.274 | 38307 | 2.771 | X-n801-k40 | 73311 | 79846 | 8.914 | 77895 | 6.253 |
| X-n251-k28 | 38684 | 39733 | 2.712 | 39404 | 1.861 | X-n819-k171 | 158121 | 169901 | 7.450 | 166419 | 5.248 |
| X-n256-k16 | 18839 | 19233 | 2.091 | 19086 | 1.311 | X-n837-k142 | 193737 | 212659 | 9.767 | 207069 | 6.881 |
| X-n261-k13 | 26558 | 27434 | 3.298 | 27200 | 2.417 | X-n856-k95 | 88965 | 93339 | 4.917 | 91904 | 3.304 |
| X-n266-k58 | 75478 | 80431 | 6.562 | 78942 | 4.589 | X-n876-k59 | 99299 | 103914 | 4.648 | 102414 | 3.137 |
| X-n270-k35 | 35291 | 35918 | 1.777 | 35726 | 1.233 | X-n895-k37 | 53860 | 57016 | 5.860 | 56042 | 4.051 |
| X-n275-k28 | 21245 | 21850 | 2.848 | 21642 | 1.869 | X-n916-k207 | 329179 | 362272 | 10.053 | 352372 | 7.046 |
| X-n280-k17 | 33503 | 34693 | 3.552 | 34273 | 2.298 | X-n936-k151 | 132715 | 146776 | 10.595 | 142537 | 7.401 |
| X-n284-k15 | 20226 | 21570 | 6.645 | 21158 | 4.608 | X-n957-k87 | 85465 | 92705 | 8.471 | 90414 | 5.791 |
| X-n289-k60 | 95151 | 99201 | 4.256 | 97797 | 2.781 | X-n979-k58 | 118976 | 128300 | 7.837 | 125484 | 5.470 |
| X-n294-k50 | 47161 | 48074 | 1.936 | 47789 | 1.332 | X-n1001-k43 | 72355 | 78832 | 8.952 | 76758 | 6.085 |
| | | | | | | Avg. Gap% | * | | 4.110 % | **2.828%** | |

