# OpenReview forum: "Hierarchical Aggregation Deconstruction Search for Vehicle Routing Problems"
_ICLR.cc/2026/Conference — Submitted to ICLR 2026_

### Official Review · Reviewer_YAT7 · 2025-10-28

**Soundness:** 2
**Presentation:** 3
**Contribution:** 3
**Rating:** 4
**Confidence:** 3

**Summary:**

This paper introduces Hierarchical Aggregation Deconstruction Search (HADES), a neural improvement method for Vehicle Routing Problems (VRPs) that addresses the limitation of existing approaches in ignoring route hierarchical structures. HADES integrates two distance-aware, anisometric positional encodings: in-route positional encoding (IPE), which captures the circular and non-uniform order of nodes within a route by leveraging cumulative travel distance, and cross-route positional encoding (CPE), which represents the global spatial relationship of each route relative to the depot and other routes.
HADES outperforms SOTA traditional OR solvers and neural methods across three VRP variants (CVRP, VRPTW, PCVRP) and multiple instance sizes (500, 1000, 2000), especially showing strong advantages in large-scale benchmarks.
I think this idea is interesting, novel and worthy of in-depth study, but the experimental setup and results are not very solid, includes the fact that the extent of the performance improvement is not significant, and the generalization ability of the new method when not trained separately on different size VRP problems has not been discussed. This makes the actual effect of this idea questionable.

**Strengths:**

1. The idea that incorporates in-route positional encoding and cross-route encoding is interesting and novel.
2. Adopts a "deconstruction-reconstruction + ASA" framework combined with winner-takes-all policy-gradient training, balancing exploration and exploitation in optimization to enhance solution quality and stability.
3. Includes comprehensive experiments with diverse baselines, multi-scale test sets, and strict time constraints, ensuring good result credibility, and authors commit to open-sourcing the code to facilitate field research.
4. The hierarchical encoding idea is model-agnostic, applicable to other similary combinatorial optimization problems.
5. Well writing and easy to understand.

**Weaknesses:**

Overall, the experimental setup and results are not very solid, including the fact that the extent of the performance improvement is not significant, and the generalization ability of the new method when not trained separately on different size VRP problems has not been discussed.
1. The comparison in Table 1 is not fair. The method proposed in this paper trains a model for each probem and each probelm size, but the other baseline methods do not do so.
2. For the ablation experiments in Tables 2 and 4, the improvement of the method proposed in this paper is very small. This indicate that the core contribution of the article is limited.

**Questions:**

I like the idea of this article(incorporates in-route positional encoding and cross-route encoding), but the current experimental results cannot fully demonstrate the effectiveness of this idea. If the author could provide additional explanations regarding the effectiveness of the idea, I would be happy to raise my rating score.
1. Additional experiment: Train a model on VRP problems with all problem sizes, and then compare the results with the baseline algorithms.
2. This article does not provide any information regarding the results of the VRP problem on a smaller scale(e.g. 50/100). So, how effective is it?
3. For cvrp problem, maybe the author could test the dataset from cvrplib.

---

> ### Author Response · Authors · 2025-11-28
>
> Thank you for highlighting the novelty, solid experiment, and good presentation of our work. We will address your concerns as follows.
>
> **`[W0 & W2]` Clarification for performance contributions**
>
> We would like to first clarify that our benchmarks operate in a *near-optimal* regime. The gaps to the strongest heuristics are already negative. In such settings, making further progress is inherently difficult. Even seemingly small improvements correspond to meaningful advances, particularly in large-scale logistics applications where fractional-percentage gains translate to substantial operational savings.
>
> However, we acknowledge your concern that the improvements may appear modest in absolute value. To better illustrate the general utility of our hierarchical anisometric encodings, we applied our hierarchical aggregation technique to  the PDP-N2S [1], which is a strong neural neighborhood search method for pickup-and-delivery (PDP) problems. We trained this model by not separately on different sizes of PDP and then evaluated the performance to further validate the generalization ability.
>
> Across all tested instance sizes, the resulting HA-N2S consistently outperforms all baselines, demonstrating non-marginal improvements in solution quality as shown in Table 4. This cross-model enhancement highlights that the hierarchical aggregation provides general representational benefits beyond our own architecture, supporting our claim that they are model-agnostic and broadly useful.
>
> You may refer to Table 4 for the results, Section 5.2 for the discussion, and Section B.5 for the experimental setting in the revised manuscript.
>
> We have also uploaded the HA-N2S code and trained checkpoint to the supplementary material, and we will make it open-source together with the HADES. We hope this would be valuable to the general NCO community.
>
> **`[W1]` Clarifaction for testing configration**
>
> Table 1 follows exactly the same evaluation protocol as NDS to ensure a fair and consistent comparison. As stated in the original NDS paper, *“for each problem and problem size, we perform a separate training run”*. This is the established practice in neural combinatorial optimization, and all learning-based baselines compared in NDS are trained under the same per-problem, per-size setting.
>
> Since our goal is to validate the effectiveness of the proposed hierarchical anisometric encodings relative to the strongest existing method, we adopt the identical setup: one model is trained for each VRP variant and each instance size, just as NDS and its neural baselines do. This ensures that the comparison in Table 1 is directly aligned with prior work and faithfully reflects the standard evaluation protocol used in the literature.

---

> ### Author Response · Authors · 2025-11-28
>
> **`[Q0]` Additional geometric analysis**
>
> Thank you for raising this idea. These were things we were planning to do but did make them in time before. We agree that explanation is important, and we have incorporated additional analysis in the revised manuscript to address this point directly.
>
> We examined the geometric patterns of the nodes selected for removal. For in-route behavior, we analyzed the normalized cumulative distance of removed nodes across 10M removals on CVRP-500. The results in Figure 5 show that the policy consistently prioritizes customers near the beginning and end of routes, which are exactly the positions where cross-route interactions and inefficient detours are more likely. This provides concrete evidence that the in-route encoding informs the model to focus on boundary regions that are most impactful for improvement.
>
> We also analyzed the angular distribution of removed nodes to study cross-route behavior. Using 200K evaluated solutions, we computed normalized angular entropy and observed that the policy concentrates removals in a small set of coherent angular sectors, whereas NDS spreads them more uniformly. This lower entropy indicates that the model learns to focus on specific geometric regions rather than applying removals indiscriminately.
>
> Together, these findings show that IPE and CPE do shape meaningful deconstruction strategies, and they provide explicit examples of the types of removals the model learns to prioritize. Thanks for your suggestion again and we believe these insights offer useful interpretability and may provide valuable hints for researchers developing future NCO models.
>
> You may refer to the Figure 5 for the results, Section 5.2 for the discussion, and Section B.6 for additional experiment results in the revised manuscript.
>
> **`[Q1]` Additional experiment cross size**
>
> Thank you for this suggestion. Our new experiment with HA-N2S follows exactly the experimental protocol of the original N2S paper, where a single model is trained on a unified distribution covering all problem sizes (21, 51, 101) and then evaluated across the same range of sizes (as detailed in the PDP-N2S paper. This setting aligns with what the reviewer is asking for, namely training one model and testing it across multiple scales.
>
> In this experiment, integrating our hierarchical anisometric encodings into N2S yields the HA-N2S variant, which consistently outperforms all baselines across every tested PDTSP and PDTSP-LIFO size. These results demonstrate that our representational design generalizes effectively across problem sizes within a single training distribution. More importantly, they validate that the benefits of our hierarchical anisometric encodings are not tied to HADES alone but provide a broadly useful mechanism that strengthens different neural improvement frameworks even under unified multi-scale training.
>
> **`[Q2]` About small scale experiment**
>
> We focus on medium- and large-scale VRP settings by following the standard evaluation protocol in recent neural improvement works. On very small instance sizes (for example, 50 or 100 nodes), all strong baselines (especially heuristics) already achieve solutions extremely close to real optimality, and the gaps between methods become marginal. As noted in prior work, these small scales do not meaningfully differentiate improvement-based approaches, nor do they reflect the practical regimes where neural methods provide the most value. Our study therefore aligns with the established practice of evaluating on problem sizes where structural differences between methods are observable and where improvement quality matters operationally. Within these settings, our method demonstrates clear gains over both neural and OR baselines.

---

> ### Author Response · Authors · 2025-11-28
>
> **`[Q3]` Additional experiment on CVRPLib**
>
> We now evaluate our method on the X-series of CVRPLib, a real-world benchmark suite that exhibits heterogeneous spatial layouts, varied customer densities, and diverse structural properties that differ substantially from the synthetic training distribution.
>
> Importantly, we directly apply the model trained on instances of size $N=500$ without any retraining or size-specific adaptation to all X instances. As reported in the revised appendix, HADES achieves an average gap of $2.828\%$, compared to NDS at $4.110\%$ under the same 60-second evaluation protocol. This improvement demonstrates that the geometric inductive biases captured by our hierarchical anisometric encodings transfer meaningfully to real benchmark distributions that differ in geometry, scale, and route topology from the training data.
>
> You may refer to Table 9 for the results on CVRPLib, and Section B.6 for the results discussion in the revised manuscript.
>
> ---
>
> Thanks again for your valuable comments, and we apologise for this late response as unexpected workload. Please let us know whether these planned experiments and analyses fully address your concerns; we remain available to make any necessary adjustments.
>
> **References**
>
> [1] Ma, Y., Li, J., Cao, Z., Song, W., Guo, H., Gong, Y. and Chee, Y.M., 2022. *Efficient neural neighborhood search for pickup and delivery problems*. arXiv preprint arXiv:2204.11399.

---

### Official Review · Reviewer_cgUz · 2025-10-31

**Soundness:** 3
**Presentation:** 3
**Contribution:** 3
**Rating:** 6
**Confidence:** 4

**Summary:**

This paper proposes HADES, a neural improvement framework for VRPs. HADES introduces a novel set of hierarchical positional encodings: 1) In-route Positional Encoding (IPE): A distance-aware sinusoidal encoding that captures the non-uniform spacing and circular, head-tail connected topology of nodes within a single route. 2) Cross-route Positional Encoding (CPE): A depot-anchored angular encoding that represents the global spatial relationship of routes relative to each other. These encodings are integrated into an encoder-decoder model that learns a deconstruction policy (i.e., which customers to remove). This policy is then used within an Augmented Simulated Annealing (ASA) search framework to iteratively improve solutions. Extensive experiments on large-scale CVRP, VRPTW, and PCVRP benchmarks show that HADES achieves state-of-the-art results, outperforming both strong Operations Research (OR) solvers and neural methods.

**Strengths:**

1. The paper is well-written and the motivation is exceptionally clear.

2. The experimental evaluation is convincing. It is benchmarked on three VRP variants (CVRP, VRPTW, PCVRP) with different scales (N=500, 1000, 2000). The comparison includes both SOTA OR solvers and learning-based methods. The authors provide detailed ablation studies, confirming that both the proposed IPE and CPE are necessary for enhanced performance.

3. This work delivers SOTA results on challenging, large-scale benchmarks. The proposed encodings may be "model-agnostic building blocks", which may not be limited to the HADES framework and could be integrated into other NCO architectures (both constructive and improvement-based methods).

**Weaknesses:**

1. The Cross-route Positional Encoding (CPE) depends on "randomly" selecting a reference route $r^*$ to define the zero-angle. This introduces a source of stochasticity that is not analyzed.

2. The CPE is "depot-anchored", calculating all route angles relative to the single depot. This is a fine assumption for the problems tackled (CVRP, VRPTW, PCVRP). However, it's unclear how this specific design would generalize to other important VRP variants, such as the Multi-Depot VRP (MDVRP). This limitation on the CPE's applicability could be discussed.

**Questions:**

1. Could the authors clarify the procedure for selecting the "randomly selected reference route" $r^*$? Is this route chosen once per instance and then fixed, or is it re-sampled during the improvement process? Have you conducted any experiments on the sensitivity of HADES to this random choice?

2. As mentioned in the weaknesses, the CPE is depot-anchored. Do the authors have any insights on how this idea of a cross-route angular encoding could be extended to problems without a single, central depot, such as Multi-Depot VRP?

---

> ### Author Response · Authors · 2025-11-28
>
> Thank you for highlighting the solid experiment design, good presentation, and strong results of our work. We will address your concerns as follows.
>
> **`[W1 & Q1]` Clarification for $r^*$ selection**
>
> Thanks for raising this valuable concern. Our CPE relies on *relative* angular relationships between routes. The choice of the reference route serves only as a phase origin for computing these relative offsets and carries no semantic meaning during each improvement step. This design follows the same principle as modern positional encoding schemes in sequence models such as GPT, Qwen and DeepSeek, where absolute indices are arbitrary and only relative structure is used by the model. Enforcing a deterministic rule might introduce a fixed geometric bias that is unrelated to the solution structure. In contrast, using a random route ensures that no artificial preference is imposed, and because the model consumes only relative angular differences, its behavior is insensitive to this choice.
>
> You may refer to Section B.6 for more $r^*$ selection discussions in the revised manuscript.
>
> **`[W2 & Q2]` Generalize to other variants**
>
> Thanks for your observation. To validate the ability of our proposed method to generalize to other important VRP variants, we have additionally examined the broader usefulness of our hierarchical anisometric encodings by integrating them into N2S [1], a strong neural neighborhood search method for pickup-and-delivery (PDP) problems.
>
> As reported in the revised manuscript, the resulting HA-N2S variant consistently outperforms all baselines across all tested PDTSP sizes. This cross-model and cross-task evidence indicates that the underlying representational design is not tied to a specific architecture and can transfer effectively to other routing formulations.
>
> You may refer to Table 4 for the HA-N2S results, Section 5.2 for the discussion, and Section B.5 for the experimental setting in the revised manuscript.
>
> We have also uploaded the HA-N2S code and trained checkpoint to the supplementary material, and we will make it open-source together with the HADES. We hope this will be valuable to the general NCO community.
>
> Regarding the multi-depot setting, our current CPE is validated and analyzed in the single-depot regime, where anchoring route angles to a unique depot naturally captures the global spatial organization of routes. Extending the encoding to multi-depot variants would require accommodating multiple anchor points. A natural direction would be to assign each depot its own positional component or to construct a multi-dimensional route-phase representation so that routes associated with different depots are encoded relative to the appropriate anchor. We agree that developing such extensions is an interesting direction for future work and a promising step toward broadening the applicability of our hierarchical aggregation design.
>
> ---
>
> Thanks again for your valuable comments, and we apologise for this late response as unexpected workload. Please let us know whether these planned experiments and analyses fully address your concerns; we remain available to make any necessary adjustments.
>
> **References**
>
> [1] Ma, Y., Li, J., Cao, Z., Song, W., Guo, H., Gong, Y. and Chee, Y.M., 2022. *Efficient neural neighborhood search for pickup and delivery problems*. arXiv preprint arXiv:2204.11399.

---

### Official Review · Reviewer_VDdB · 2025-11-01

**Soundness:** 2
**Presentation:** 3
**Contribution:** 2
**Rating:** 4
**Confidence:** 4

**Summary:**

This paper introduces HADES (Hierarchical Aggregation Deconstruction Search), a neural improvement method for VRPs that incorporates distance-aware, anisometric positional encodings. The method features two complementary components: (1) in-route positional encoding (IPE) that captures circular and non-uniform ordering within tours using cumulative travel distance, and (2) cross-route positional encoding (CPE) that represents route membership via depot-anchored angular encoding. The approach is evaluated on CVRP, VRPTW, and PCVRP, demonstrating improvements over strong baselines.

**Strengths:**

- The insight that VRP solutions require anisometric (distance-aware) rather than isometric (index-based) positional encodings is compelling. The combination of IPE (within-route) and CPE (across-route) encodings elegantly captures the hierarchical structure of VRP solutions, addressing a genuine gap in how neural methods represent solutions.
- HADES achieves consistent improvements over HGS/PyVRP-HGS baselines (-0.30% to -4.94% gaps) and outperforms the recent NDS method across CVRP, VRPTW, and PCVRP.
- The paper is well-written with effective figures (especially Figure 3 contrasting LLM vs. VRP positional encodings).

**Weaknesses:**

- On CVRP-500, HADES achieves only -0.30% improvement over HGS, barely better than NDS (-0.20%). While larger instances show stronger gains, the marginal improvement on smaller scales suggests the method's advantage is primarily in scalability rather than fundamental quality.
- Incomplete Analysis of Discovered Heuristics: The paper claims operators are "novel and powerful" but provides no concrete examples. No visualization or interpretation of what the model learns through IPE/CPE. Missing analysis of which types of removals the model prioritizes (e.g., does it preferentially select customers with high cumulative distance? customers at angular boundaries?
- Computational Cost Not Reported: Training cost is not mentioned (2000 epochs × 150K instances × 128 rollouts is substantial). Comparison of training cost vs. performance gain is missing. No discussion of whether the improvements justify the training investment.
- Methodological Concerns: CPE reference route selection: Randomly selecting a reference route r* seems arbitrary. How sensitive is performance to this choice? Would a consistent selection (e.g., leftmost route) be more stable? Head-tail alignment: The circular normalization ˆdi = (di/dL) · 2π ensures IPE(v1) = IPE(vL), but this assumes symmetric distance matrices. For asymmetric problems, this alignment may be misleading.
- The term "anisometric" is used extensively but never formally defined. A brief definition would help readability.
- Algorithm notation inconsistency: the paper uses both πθ(·|·) and p(·|·) for the policy
- Experimental Design: Why use different numbers of augmentations for different problems (8 for CVRP/VRPTW, 128 for PCVRP)? This inconsistency is not explained. Test set sizes vary (128 for CVRP-500 from Drakulic, 100 for CVRP-1000/2000 from Ye, 250 new for VRPTW/PCVRP). This heterogeneity complicates comparison. No error bars or variance across runs reported.
- NCO-LLM and other LLM-based methods are included but not discussed in detail.
- Generalization Claims: Table 3 tests low-capacity and clustered distributions, but these are relatively mild distribution shifts. More challenging shifts (e.g., extremely heterogeneous demands, mixed urban/rural layouts) would strengthen the generalization story.

**Questions:**

1. Computational Cost Analysis
What is the total training cost (GPU hours, wall-clock time, dollars) for each problem/size? On CVRP-500, you improve by only 0.10% absolute (36.66 → 36.60) over HGS. Does the performance gain justify the training investment, and at what deployment scale does HADES become cost-effective?

2. IPE Normalization Across Routes
You normalize cumulative distance per route: ˆdi = (di/dL) · 2π. How does this handle tours of vastly different lengths in the same solution (e.g., one route of length 100, another of length 10)? Doesn't the same physical distance map to different IPE values across routes, breaking the "distance-aware" property you emphasize? Have you tried global normalization instead?

3. CPE Reference Route Selection
The reference route r is randomly selected, meaning different random seeds produce different CPE values for the same solution. How does the model handle this non-determinism during training and testing? Have you compared random vs. deterministic selection (e.g., leftmost route)? Additionally, why not use circular mean instead of arithmetic mean for ϕr to avoid wraparound issues (e.g., averaging 10° and 350° gives 180°)?

4. Interpretability and Learned Strategies
Can you provide concrete examples with visualizations showing what removal patterns HADES learns? Does it preferentially remove customers with high cumulative distance, customers at route boundaries, or customers from specific angular sectors? Can you visualize decoder attention weights to show which customers receive high attention? This analysis is crucial for understanding why your method works and building trust for practical deployment.

5. Statistical Rigor
Table 1 reports point estimates with no error bars or variance. Can you provide standard deviations across multiple runs, confidence intervals, and statistical significance tests (e.g., paired t-test vs. HGS)?

6. Ablation to Isolate Contributions
The IPE ablation shows only 0.02% improvement (36.56 → 36.54), while CPE shows 0.04% (36.57 → 36.54). These gains seem marginal. Have you tested an "NDS + IPE/CPE" configuration to isolate whether the improvements come from the encodings vs. other algorithmic choices (winner-takes-all training, architecture, hyperparameters)? Can you attribute the 0.10-0.14% gain over NDS to specific components?

7. Generalization Beyond Training Distribution
You train on N ≤ 400 but test on N = 2000 (5× larger). Why not train on larger instances? Table 3 tests relatively mild distribution shifts (low capacity, clustered layouts). Can you test more challenging shifts like extreme demand heterogeneity (demand ∈ {1, 100}), mixed urban/rural layouts, asymmetric distances, or multi-depot problems? How well do the circular and radial inductive biases hold in these cases?

8. Experimental Design Concerns
Several design choices seem inconsistent or unexplained: (a) Why use 8× augmentations for CVRP/VRPTW but 128× for PCVRP? (b) Your test protocol uses 200 rollouts × 8 augmentations = 1600 evaluations while HGS/SISRs use deterministic search. Is this a fair computational comparison? (d) Why use different test set sizes (128, 100, 250) across benchmarks?

9. Practical Deployment and Future Directions
Can HADES handle real-world constraints like driver breaks, multiple depots, and heterogeneous fleets, or would you need to retrain for each constraint?

**Minor Corrections**

- Line 229: "ˆdi = di/dL · 2π" should clarify whether this is element-wise
- Figure 4: The illustration could benefit from showing actual angle values

---

> ### Author Response · Authors · 2025-11-28
>
> Thank you for highlighting the novelty, solid results, and good presentation of our work. We will address your concerns as follows.
>
> **`[W1 & Q6]` Clarification on performance contribution**
>
> We would like to first clarify that CVRP-500 lies in an extremely competitive regime where both HADES and NDS already achieve *negative* gaps relative to HGS. In such near-optimal settings, further improvements are inherently difficult. Even seemingly small improvements correspond to meaningful advances, particularly in large-scale logistics applications where fractional-percentage gains translate to substantial operational savings. We therefore view this margin as meaningful within the tight performance region where even small advances reflect solid improvements in solution refinement.
>
> However, we acknowledge your concern that the improvements within our own architecture may appear modest in absolute value. To better illustrate the general utility of our hierarchical anisometric encodings, we applied our hierarchical aggregation technique to the PDP-N2S [1], which is a strong neural neighborhood search method for pickup-and-delivery (PDP) problems. The results are updated to the revised manuscript. Across all tested instance sizes, the resulting HA-N2S variant consistently outperforms all baselines, demonstrating non-marginal improvements in solution quality as shown in Table 4. This cross-model and cross-task evidence shows that the advantage of our hierarchical anisometric encodings is not limited to scalability, but rather provides a generally useful representational mechanism.
>
> You may refer to Table 4 for the results, Section 5.2 for the discussion, and Section B.5 for the experimental setting in the revised manuscript.
>
> We have also uploaded the HA-N2S code and trained checkpoint to the supplementary material, and we will make it open-source together with the HADES. We hope this will be valuable to the general NCO community.
>
> **`[W2 & Q4]` Additional geometric analysis**
>
> Thank you for pointing out the analysis of what the model learns through IPE and CPE. These were things we were planning to do, but didn't do them in time before. We agree that interpretability is important, and we have incorporated additional analysis in the revised manuscript to address this point directly.
>
> We examined the geometric patterns of the nodes selected for removal. For in-route behavior, we analyzed the normalized cumulative distance of removed nodes across 10M removals on CVRP-500. The results in Figure 5 show that the policy consistently prioritizes customers near the beginning and end of routes, which are exactly the positions where cross-route interactions and inefficient detours are more likely. This provides concrete evidence that the in-route encoding informs the model to focus on boundary regions that are most impactful for improvement.
>
> We also analyzed the angular distribution of removed nodes to study cross-route behavior. Using 200K evaluated solutions, we computed normalized angular entropy and observed that the policy concentrates removals in a small set of coherent angular sectors, whereas NDS spreads them more uniformly. This lower entropy indicates that the model learns to focus on specific geometric regions rather than applying removals indiscriminately.
>
> Together, these findings show that IPE and CPE do shape meaningful deconstruction strategies, and they provide explicit examples of the types of removals the model learns to prioritize. Thanks for your suggestion again, and we believe these insights offer useful interpretability and may provide valuable hints for researchers developing future NCO models.
>
> You may refer to Figure 5 for the results, Section 5.2 for the discussion, and Section B.6 for additional experiment results in the revised manuscript.

---

> ### Author Response · Authors · 2025-11-28
>
> **`[W3]` Clarification for computational cost**
>
> HADES is trained under the same configuration as NDS for the fairness, processing 150K instances per epoch with 128 rollouts as shown on page 7. In practice, this results in a training time of roughly 9–11 days on a single A100 GPU, comparable to prior neural deconstruction frameworks rather than exceeding typical costs in this line of work.
>
> Importantly, this training is performed only once, while inference is extremely efficient. During evaluation, HADES operates under the same per-instance wall-clock budgets as all baselines (60/120/240 seconds) and achieves the best overall solution quality across CVRP, VRPTW, and PCVRP, demonstrating that the practical runtime bottleneck lies in inference rather than training.
>
> We also emphasize that even sub-percent improvements are meaningful for real-world VRPs, where small reductions in route length accumulate into substantial long-term savings in operating cost and emissions. At the largest scale, HADES improves gaps by up to 2.45% on CVRP, 3.84% on VRPTW, and 4.94% on PCVRP compared with strong OR solvers, while using the same inference-time budget. These gains outweigh the one-time training investment and reflect that the model’s hierarchical positional design leads to more effective deconstruction within identical runtime limits.
>
> You may refer to Section B.4 for more testing configuration details in the revised manuscript.
>
> **`[W4.1 & Q3]` Clarification for $r^*$ selection**
>
> Thanks for your careful examination of the positional encoding design and welcome the opportunity to clarify these methodological choices. Our CPE relies on relative angular relationships between routes. The choice of the reference route serves only as a phase origin for computing these relative offsets and carries no semantic meaning. This design follows the same principle as modern positional encoding schemes in sequence models such as GPT, Qwen and DeepSeek, where absolute indices are arbitrary and only relative structure is used by the model. Enforcing a deterministic rule might introduce a fixed geometric bias that is unrelated to the solution structure. In contrast, using a random route ensures that no artificial preference is imposed, and because the model consumes only relative angular differences, its behavior is insensitive to this choice.
>
> Regarding the computation of the representative angle for each route, we use a simple aggregation method that first identifies a continuous angular interval covering all node angles in the route (effectively unwrapping the circular coordinates locally) and then averages within that interval. This prevents wraparound artifacts such as mixing angles near 0 and 2π. Since the model consumes only these relative angular relationships, the behavior of the policy is insensitive to the particular aggregation scheme.
>
> You may refer to Section B.6 for more $r^*$ selection discussions in the revised manuscript.
>
> **`[W4.2]` About asymmetric problems**
>
> We agree that asymmetric VRPs are important in practice. One possible direction for handling non-reversible costs is to extend the positional representation with directional components that capture forward and backward travel characteristics. As our work is the first to investigate distance- and angle-aware positional encodings within a hierarchical aggregation framework for VRPs, we view dedicated exploration of asymmetric variants as a valuable next step.
>
> **`[W5]` Definition  for "anisometric"**
>
> We have the clarification definition about the "anisometric" in Section A.3 in the revised manuscript. Thanks for helping us to make the paper easier to read.
>
> **`[W6]` Fix policy notation**
>
> Thanks for pointing out the problem. We have revised the notation usage in Equation 1 in the manuscript to use a consistent notation throughout.

---

> ### Author Response · Authors · 2025-11-28
>
> **`[W7.1 & Q8]` Clarification for experimental design**
>
> Regarding the number of augmentations, our settings follow the established configuration of NDS to maintain fairness in comparison, as the augmentation count interacts with the search horizon and acceptance dynamics in a way that is solver-specific. Using the same configuration as NDS ensures that the comparison remains consistent with prior work and avoids introducing confounding changes to the search behavior.
>
> On the test sets, we note that all datasets used in our experiments are standard and widely adopted in the VRP literature. Thus we use the established generators from prior works. Using these canonical datasets is important for comparability and faithfully aligns our evaluation with the benchmarks used by existing neural and OR baselines.
>
> **`[W7.2 & Q5]` Information for running variance**
>
> In the revised manuscript, we now include the mean and standard deviation of our method across three independently trained models for each setting. These results are provided in Table 8 of the appendix. Even though baseline methods do not report variance or multi-run statistics in their original papers, since all approaches are evaluated on the same standardized test datasets under identical runtime and hardware conditions, we believe this remains a fair and consistent comparison aligned with existing evaluation practices in neural improvement and VRP research.
>
> You may refer to Table 8 for the running variance results, Section B.4 for more testing configurations in the revised manuscript.
>
> **`[W8]` Information for NCO-LLM**
>
> We have expanded the description of the NCO-LLM in the appendix and clarified its role in the main comparison. NCO-LLM is a strong neural construction framework built on LEHD, enhanced by LLM-designed heuristics for logit reshaping during decoding. We include it as a representative method from the emerging line of LLM-assisted neural routing solvers, alongside recent approaches such as ReEvo-ACO and LLM-driven large neighborhood search, which similarly employ LLMs to design or refine heuristic components.
>
> Our intention in including NCO-LLM is to position our method against the strongest available learning-based baselines, including those augmented with LLM-guided heuristics. As reported in the main results, our method achieves lower objective values than NCO-LLM across all CVRP sizes under identical time budgets, demonstrating that a dedicated improvement framework with hierarchical positional encodings remains highly competitive even when compared to models benefiting from LLM-derived heuristic guidance. We believe this comparison further reinforces the effectiveness of our approach and highlights that explicit modeling of route structure through anisometric positional encodings can offer advantages complementary to, and in some cases exceeding, those provided by LLM-based heuristic design.
>
> You may refer to Section B.6 for more discussions about the NCO-LLM in the revised manuscript.
>
> **`[W9 & Q7]` Additional generalization experiment**
>
> In addition to the controlled distribution-shift experiments reported in Table 3, we now evaluate our method on the X-series of CVRPLib, a real-world benchmark suite that exhibits heterogeneous spatial layouts, varied customer densities, and diverse structural properties that differ substantially from the synthetic training distribution.
>
> Importantly, we directly apply the model trained on instances of size $N=500$ without any retraining or size-specific adaptation to all X instances. As reported in the revised appendix, HADES achieves an average gap of $2.828\%$, compared to NDS at $4.110\%$ under the same 60-second evaluation protocol. This improvement demonstrates that the geometric inductive biases captured by our hierarchical anisometric encodings transfer meaningfully to real benchmark distributions that differ in geometry, scale, and route topology from the training data.
>
> Beyond the empirical results on CVRPLib, we would also like to highlight that our method is model-agnostic: the proposed IPE and CPE modules can be plugged into any improvement-based neural framework. In the revised manuscript, we demonstrate this by applying our hierarchical aggregation design to the N2S model, yielding consistent improvements across PDTSP variants. This cross-model evidence shows that the underlying representational principles generalize beyond a specific architecture and support the broader robustness claims of our approach. Together, these results provide a stronger validation of HADES under more realistic and diverse distribution shifts.
>
> You may refer to Table 9 for the results on CVRPLib, and Section B.6 for the results discussion in the revised manuscript.

---

> ### Author Response · Authors · 2025-11-28
>
> **`[Q1]` Computational cost analysis**
>
> For CVRP-500, as we mentioned in the previous point, we note that this benchmark lies in an extremely competitive regime where both HADES and NDS already achieve negative gaps relative to HGS. In such near-optimal settings, further improvements are intrinsically difficult, and even fractional-percentage gains can translate into meaningful cumulative savings in real-world logistics. The improvement therefore reflects a genuine advance within a saturated benchmark rather than statistical noise.
>
> Regarding training cost, HADES follows the same configuration as NDS for fairness, and most importantly, this cost is incurred only once, while test-time inference is highly efficient and operates under identical wall-clock budgets as all baselines. Since deployment bottlenecks arise in inference rather than training, the one-time training investment is amortized across repeated use and is comparable to existing neural deconstruction frameworks.
>
> **`[Q2]` About IPE normalization**
>
> The IPE is intended to encode the relative position of each node within its own tour rather than absolute metric distances across different tours. This choice reflects the structure of VRP solutions: each route is an independent closed cycle, and the search policy’s decision for removing a node depends primarily on its position within that cycle (for example, proximity to predecessors and successors, position along the traversal order) rather than on absolute distance comparisons across different routes.
>
> Normalizing each route independently ensures that the encoding preserves the circular topology of every tour, including the natural property that the first and last nodes (i.e. depot) share the same positional representation, which would not hold under a global normalization scheme.
>
> Importantly, the distance-awareness we aim to capture is local and route-specific: IPE communicates how far along the current route a node lies, relative to other nodes in the same tour. This is exactly the information required for deconstruction, as shown in our rollout visualizations. Under a global normalization, short routes would be compressed into small angular spans while long routes would dominate, causing nodes with similar roles in different routes to receive disproportionately different encodings and potentially biasing the policy toward or against particular routes for reasons unrelated to their structural roles.
>
> Finally, we note that in high-quality VRP solutions, route lengths do not typically diverge arbitrarily due to load constraints and efficiency considerations, so per-route normalization remains stable in practice. While extending IPE to support additional global signals could be an interesting direction, our experiments indicate that the current route-wise formulation provides effective and robust positional structure for learning deconstruction policies.
>
> **`[Q7]` Clarification for generalization setting**
>
> Thank you for raising this question. We clarify that our models are not trained only on instances with N ≤ 400. As described in Section 5.1 of the manuscript, we follow the experimental protocol of prior work and train a separate policy for each problem size, including N = 2000.
>
> **`[Q9]` Future directions about real-world setting**
>
> Thanks for your question on broader applicability. HADES is a neural improvement framework tailored to standard VRP settings rather than a universal foundation model. Incorporating additional real-world constraints such as driver breaks, multiple depots, or heterogeneous fleets would follow the usual practice in the NCO literature, where models are adapted or retrained to match the target problem. Developing a unified model that can accommodate many constraint regimes without retraining is an interesting direction. We believe that this is a promising direction for future research toward more general-purpose learned routing solvers.
>
> **Minos**
>
> Thanks for the suggestions, we have emphasized the element-wise definition for $d$ in IPE in the revised manuscript. About the Figure 4, we treat it as an intuitive illustration for the CPE. For actual angle values, the audience could now refer to the new Figure 5 for better understanding.
>
> ---
>
> Thanks again for your valuable comments, and we apologise for this late response as unexpected workload. Please let us know whether these planned experiments and analyses fully address your concerns; we remain available to make any necessary adjustments.
>
> [1] Ma, Y., Li, J., Cao, Z., Song, W., Guo, H., Gong, Y. and Chee, Y.M., 2022. *Efficient neural neighborhood search for pickup and delivery problems*. arXiv preprint arXiv:2204.11399.

---

### Official Review · Reviewer_wb1W · 2025-11-02

**Soundness:** 2
**Presentation:** 3
**Contribution:** 1
**Rating:** 2
**Confidence:** 3

**Summary:**

This paper proposes Hierarchical Aggregation Deconstruction Search (HADES), a neural improvement framework for solving Vehicle Routing Problems (VRPs) under the umbrella of Neural Combinatorial Optimization (NCO). The core motivation is to address the limitation of existing neural improvement methods, which fail to capture the hierarchical structure and anisometric nature of VRP solutions. HADES introduces two complementary positional encodings: (1) In-Route Positional Encoding (IPE), which uses cumulative travel distance to model the circular and non-uniform order of nodes within a route; and (2) Cross-Route Positional Encoding (CPE), which leverages depot-anchored average angles of routes to capture global inter-route structural relations.

**Strengths:**

1. HADES explicitly targets two under-explored properties of VRP solutions: hierarchical structure and anisometry. This design aligns with the intrinsic geometric characteristics of VRPs.
2. The authors conduct extensive experiments across multiple dimensions: (1) three canonical VRP variants to validate generality; (2) large-scale instances to test scalability; (3) both OR solvers and neural methods as baselines to ensure competitiveness.

**Weaknesses:**

1. The core value of IPE and CPE is undermined by their minimal performance contributions.
2. The authors claim IPE and CPE are “grounded in VRP geometry,” but provide no rigorous theoretical support for key design choices.

**Questions:**

According to the ablation experiment, IPE and CPE, the core contributions of the paper, do not seem to play a great role. How to view the importance of the method?

---

> ### Author Response · Authors · 2025-11-28
>
> Thank you for highlighting the novelty, solid experiment, and good presentation of our work. We will address your concerns as follows.
>
> **`[W1]` Clarification for performance contributions**
>
> We would like to first clarify that our benchmarks operate in a *near-optimal* regime. The gaps to the strongest heuristics are already negative. In such settings, making further progress is inherently difficult. Even seemingly small improvements correspond to meaningful advances, particularly in large-scale logistics applications where fractional-percentage gains translate to substantial operational savings.
>
> However, we acknowledge your concern that the improvements may appear modest in absolute value. To better illustrate the general utility of our hierarchical anisometric encodings, we applied our hierarchical aggregation technique to  the PDP-N2S [1], which is a strong neural neighborhood search method for pickup-and-delivery (PDP) problems. Across all tested instance sizes, the resulting HA-N2S consistently outperforms all baselines, demonstrating non-marginal improvements in solution quality as shown in Table 4. This cross-model enhancement highlights that the hierarchical aggregation provides general representational benefits beyond our own architecture, supporting our claim that they are model-agnostic and broadly useful.
>
> You may refer to Table 4 for the results, Section 5.2 for the discussion, and Section B.5 for the experimental setting in the revised manuscript.
>
> We have also uploaded the HA-N2S code and trained checkpoint to the supplementary material, and we will make it open-source together with the HADES. We hope this will be valuable to the general NCO community.
>
> **`[W2]` Additional geometric analysis**
>
> We would like to first clarify that we do not claim a theoretical contribution; our use of the word “grounded” was intended to mean that the positional encodings are principled and informed by established ideas in geometric deep learning and VRP structure. To avoid confusion, we are willing to change this phrasing to “designed to better represent the solution structure.”
>
> Our designs are directly motivated by observable geometric properties of VRP solutions: IPE uses cumulative travel distance along a route and enforces head–tail consistency, while CPE is based on the depot-centered angular distribution of routes, as described in Section 4.1 and Figure 4. These choices are not arbitrary but reflect common structural patterns in near-optimal routing solutions.
>
> To further support these design choices, we provided a new statistical geometric analysis. The results in Figure 5 show that HADES preferentially removes nodes near route boundaries, where cross-route interactions are most pronounced, and focuses on coherent angular sectors rather than uniformly distributing removals. This behavior aligns with the geometric intuition behind both encodings and demonstrates that the model effectively uses the intended signals. We believe that these analyses supply evidence that the encodings match meaningful VRP geometry and contribute to improved deconstruction decisions. And these observations also offer useful hints for the broader NCO community, as they highlight geometric patterns that future models and heuristics may explicitly exploit.
>
> You may refer to Figure 5 for the results, Section 5.2 for the discussion, and Section B.6 for additional experiment results in the revised manuscript.

---

> ### Author Response · Authors · 2025-11-28
>
> **`[Q]` Clarification for the IPE, CPE importance**
>
> As mentioned in the response to `[W1]`, our experiments operate in an extremely competitive, near-optimal region. Althogh further improvements are inherently difficult, improvements of IPE and CPE are consistent and appear precisely in the region. We therefore view these results as evidence that both components contribute effectively to refining solutions within a high-performance regime.
>
> We also acknowledge your concern that improvements within our own architecture may understate the general importance of the method. In the experiment of HA-N2S. The results show that our hierarchical aggregation design consistently improves over the original N2S across all evaluated instance sizes. This cross-model and cross-task evidence supports two key points: the encodings offer benefits beyond HADES, and the mechanism is indeed model-agnostic, as discussed in the "Model-agnostic generalization" paragraph in the revised manuscript. We believe this broader applicability highlights the importance of the proposed hierarchical anisometric encodings and provides useful insight for the NCO community.
>
> ---
>
> Thanks again for your valuable comments, and we apologise for this late response as unexpected workload. Please let us know whether these planned experiments and analyses fully address your concerns; we remain available to make any necessary adjustments.
>
> **References**
>
> [1] Ma, Y., Li, J., Cao, Z., Song, W., Guo, H., Gong, Y. and Chee, Y.M., 2022. *Efficient neural neighborhood search for pickup and delivery problems*. arXiv preprint arXiv:2204.11399.

---

### Author Response · Authors · 2025-12-03
**Final Remark**

Dear ACs and reviewers,

We sincerely thank the previous and current ACs for their service and all reviewers for their valuable comments and constructive suggestions throughout the review process. We appreciate the time and effort invested in providing detailed feedback that has helped strengthen our manuscript.

We thank reviewers for acknowledging that our paper presents a *well-motivated* and *novel* approach to neural improvement methods for VRP (`wb1W`, `VDdB`, `cgUz`, `YAT7`). The insight that VRP solutions require anisometric, distance-aware positional encodings rather than isometric index-based encodings is recognized as *compelling* (`VDdB`). Our experimental evaluation is *comprehensive* and *convincing*, covering three VRP variants (CVRP, VRPTW, PCVRP) across multiple scales with comparisons to both strong OR solvers and neural baselines (`cgUz`, `VDdB`). Our manuscript is also acknowledged as *well-written* with clear motivation and effective figures (`wb1W`, `VDdB`, `cgUz`, `YAT7`).

During the rebuttal, we have addressed the following primary concerns raised by reviewers:

1. **Clear performance contribution** (`wb1W`, `VDdB`, `YAT7`). We clarified that our benchmarks operate in a *near-optimal* regime where gaps to the strongest heuristics are already negative, making further improvements inherently difficult. During the rebuttal, we demonstrated the general utility of our hierarchical anisometric encodings by integrating them into a strong neural neighborhood search method for PDP problems. The results demonstrate non-marginal improvements and confirm that our encodings are model-agnostic and broadly useful.

2. **New geometric findings** (`VDdB`, `YAT7`). We provided new statistical geometric analyses of the learned deconstruction strategy. Our findings show that IPE and CPE do shape meaningful deconstruction strategies, and they provide explicit examples of the types of removals the model learns to prioritize. We believe these insights offer useful interpretability and may provide valuable hints for researchers developing future NCO models.

3. **Additional generalization** (`VDdB`, `cgUz`, `YAT7`). We evaluated our method on CVRPLib, a real-world benchmark suite with heterogeneous spatial layouts. Without any retraining, HADES achieves an average gap of 2.828% compared to NDS at 4.110% under the same evaluation protocol. Furthermore, the HA-N2S experiment follows the original N2S protocol, where a single model is trained on a unified distribution covering all problem sizes and then evaluated across the same range, demonstrating that our encodings generalize effectively across problem sizes within a single training distribution.

4. **Discussion on CPE angle calculation** (`VDdB`, `cvUz`). We clarified that the reference route selection serves as a phase origin for computing *relative* angular offsets. This design follows the same principle as modern positional encoding schemes in sequence models such as GPT, Qwen, and DeepSeek. Using a random route ensures that no artificial geometric bias is imposed.

5. **More experimental details** (`VDdB`, `YAT7`). We provided running variance across three independently trained models for all settings. Regarding experimental design choices, such as augmentation counts and test set sizes, we clarified that our settings follow the established NDS configuration and use standard datasets from prior work to maintain fairness and comparability with existing baselines.

We have updated the manuscript to include: (1) the HA-N2S experiment demonstrating model-agnostic generalization (Table 4, Section 5.2, Section B.5); (2) geometric analysis of the learned deconstruction strategy (Figure 5, Section 5.2, Section B.6); (3) evaluation on CVRPLib X-series instances (Table 9, Section B.7); (4) running variance across multiple training runs (Table 8, Section B.4); (5) clarifications on the anisometric definition (Section A.3), CPE reference route selection, and NCO-LLM baseline description. The related code for new experiments during rebuttal is also updated in the supplementary materials.

We believe HADES advances the state of the art in neural improvement methods for VRPs by introducing the first hierarchical anisometric positional encodings tailored to routing solutions, providing both strong empirical results and interpretable geometric insights that can benefit future research in the NCO community. We respectfully ask the AC to consider these major improvements in their final decision.

Best regards,

HADES's authors.

---

### Meta-Review · Area_Chair_8Hna · 2025-12-29

**Summary:**

This paper proposes HADES, a neural improvement framework for VRPs. HADES introduces a novel set of hierarchical positional encodings: 1) In-route Positional Encoding (IPE): A distance-aware sinusoidal encoding that captures the non-uniform spacing and circular, head-tail connected topology of nodes within a single route. 2) Cross-route Positional Encoding (CPE): A depot-anchored angular encoding that represents the global spatial relationship of routes relative to each other.

Reviewers post concerns on experiments and novelty, which are valid and insightful. Based on the current manuscript, I tend to reject this paper for 1) the marginal theoretical and experimental support for their main contribution, CPE and IPE. 2) marginal improvement to the baseline NDS. 3) Abstract illustration of the motivation is not widely recognized by reviewers.

As for my concern about the proposed idea, the proposed IPE idea is quite novel, but the idea of representing the location information (as mentioned ``Where the route is located?``) within positional encodings has been widely discussed, e.g., [1] uses the retary positional encoding idea in designing positional enbeddings.

[1] Zheng, Zhi, et al. "Dpn: Decoupling partition and navigation for neural solvers of min-max vehicle routing problems." arXiv preprint arXiv:2405.17272 (2024).

**Reviewer Concerns:**

Authors have solved reviewers' concerns about experiments like CVRPLib. They also provide more explanation on their settings and designs.

**Reviewer Scores:**

Probably keep negative evaluations.

---

### Decision · Program_Chairs · 2026-01-26

Reject